# On the Learnability of Test-Time Adaptation: A Recovery Complexity Perspective

Zhi Zhou [1]   Ming Yang [1 2]   Shi-Yu Tian [1 2]   Kun-Yang Yu [1 2]   Lan-Zhe Guo [1 3]   Yu-Feng Li [1 2]

## Abstract

Test-time adaptation (TTA) aims to adapt models to maintain reliable performance on non-stationary test streams without requiring labeled data. Despite its empirical success, the learnability of TTA under non-stationary streams remains unexplored. A key challenge is the lack of a principled theoretical framework that simultaneously aligns with the TTA objective and captures both continuously evolving distribution shifts and intrinsic information constraints. To address this gap, we propose the first theoretical framework for studying the learnability of TTA and introduce $(\epsilon, \delta)$-Recovery Complexity and $(\epsilon, \rho)$-TTA Learnability. Recovery complexity measures the post-shift time needed to maintain excess risk below a target level with high probability, and is further extended to TTA learnability, which measures the long-term reliability of TTA. Within this framework, we introduce a novel discrete surrogate for non-stationary test streams, enabling a unified and tractable analysis of both gradual and abrupt shifts. We derive order-wise matching lower and upper bounds on recovery complexity, revealing fundamental limits of TTA and an intrinsic adaptivity-information trade-off. These results provide unified learnability guarantees for TTA that complement regret-based analyses.

## 1. Introduction

Deep learning models (He et al., 2016) often suffer significant performance degradation when encountering distribution shifts (Gulrajani & Lopez-Paz, 2021). As a promising remedy, test-time adaptation (TTA) has emerged as a novel paradigm that aims to adapt models to maintain reliable performance using only unlabeled test data, and has demonstrated empirical success across a wide range of domains, including computer vision (Wang et al., 2021; 2022), tabular modeling (Zhou et al., 2025a; He & Shi, 2026; Ren et al., 2024), and natural language processing (Liu et al., 2025). Despite this progress, recent studies (Zhao et al., 2023; Niu et al., 2023; Zhou et al., 2023) have shown that TTA methods can fail under complex distribution shifts, motivating a theoretical understanding of when TTA can remain reliable over non-stationary test streams.

Although there exist theoretical studies for learning with streams (Shalev-Shwartz, 2012; Zhang et al., 2023), they are not fully compatible with TTA. Online learning (Bottou, 1999; Shalev-Shwartz, 2012; Zhang et al., 2018b; Zhao et al., 2024) typically evaluates algorithms through regret against fixed or time-varying comparators, emphasizing cumulative performance rather than the post-shift recovery and instantaneous reliability required by TTA. Several TTA studies also provide theoretical analyses. AdaNPC (Zhang et al., 2023) derives performance bounds for memory-based non-parametric classifier, while ATTA (Gui et al., 2024) studies adaptation in settings where labels can be actively acquired. Their assumptions on the algorithms or label availability limit their applicability to real-world TTA problems.

These limitations highlight a fundamental gap between existing theoretical results and the requirements of TTA under unlabeled non-stationary test streams. Specifically, (i) TTA emphasizes instantaneous reliability, focusing on how quickly a model can recover and maintain satisfactory performance after a distribution shift; and (ii) the information constraints imposed by unlabeled data, potentially misaligned surrogate supervision, and complex non-stationarity are not adequately captured by existing learning frameworks. Together, these challenges call for a novel framework that introduces TTA-specific measures and captures non-stationarity.

In this work, we propose the first principled theoretical framework for studying the learnability of test-time adaptation. Specifically, we develop a unified model for complex non-stationary test streams by combining a Wasserstein-quantized surrogate for distribution shifts in Assumption 2.4

[1]State Key Laboratory of Novel Software Technology, Nanjing University, China [2]School of Artificial Intelligence, Nanjing University, China [3]School of Intelligence Science and Technology, Nanjing University, China. Correspondence to: Yu-Feng Li <liyf@nju.edu.cn>.

*Proceedings of the 43$^{rd}$ International Conference on Machine Learning*, Seoul, South Korea. PMLR 306, 2026. Copyright 2026 by the author(s).

with a $\phi$-mixing model for temporal dependence in Assumption 2.7. This stream model captures both global distributional evolution and local temporal dependence, enabling information-theoretic and optimization-based analyses of TTA. Building on this framework, we introduce the notion of $(\epsilon, \delta)$-*Recovery Complexity*, which quantifies the post-shift time needed for a TTA algorithm to maintain excess risk below a target level with high probability. Then, we establish a formal transfer from $(\epsilon, \delta)$-*Recovery Complexity* to $(\epsilon, \rho)$-*TTA Learnability*, allowing us to study the long-term reliability of TTA algorithms over entire non-stationary data streams. The resulting order-wise matching problem-level lower bound and algorithm-level upper bound on recovery complexity reveal intrinsic information limits of TTA and a fundamental trade-off between recovery speed and target excess risk. Finally, we connect our $(\epsilon, \rho)$-*TTA Learnability* to dynamic regret, clarifying both the relationship and the fundamental differences between our framework and regret-based analyses in non-stationary online learning. The overall illustration of this paper is shown in Figure 1. To summarize, the contribution of this paper is as follows:

1. We propose the first principled theoretical framework for studying the learnability of test-time adaptation, introducing $(\epsilon, \delta)$-*Recovery Complexity* and $(\epsilon, \rho)$-*TTA Learnability* to measure TTA from local post-shift and global stream-level perspectives.

2. We develop a unified model for complex non-stationary test streams by combining a Wasserstein-quantized surrogate for distribution shifts with a $\phi$-mixing model for temporal dependence, enabling tractable analyses.

3. We derive an order-wise matching problem-level minimax lower bound and algorithm-level upper bound on recovery complexity, revealing intrinsic limits and the fundamental trade-off of TTA.

4. We establish a formal connection between our $(\epsilon, \rho)$-*TTA Learnability* and dynamic regret, elucidating both their relationship and differences between our framework and existing regret-based analyses.

## 2. Test-Time Adaptation Formulation

In this section, we provide the formulation of test-time adaptation problem, including assumptions on the proxy and task losses, the model of the non-stationary test stream, and a competitive objective. Our formulation aims to establish a clear foundation for the subsequent analysis.

### 2.1. Test-Time Adaptation Problem

Let $\mathcal{X} \subseteq \mathbb{R}^d$ denote the input space and $\mathcal{Y} = \{1, \ldots, K\}$ the label space, where $d$ is the input dimensionality and $K$ is the number of classes. We consider a model $f_{\theta_1} : \mathcal{X} \to \mathcal{Y}$

pre-trained on the training distribution, with $\theta_1 \in \Theta$ denoting the initial model parameters. During the testing phase, the model is deployed to a non-stationary test data stream $\mathcal{S} = \{D_t\}_{t=1}^T$, where each batch $D_t$ contains samples drawn from an underlying joint data distribution $\mathcal{P}_t$ on $\mathcal{X} \times \mathcal{Y}$ at time $t$. Here, we write $\mathcal{P}_{t,X}$ for the input marginal of $\mathcal{P}_t$. At each time step $t$, the current model $f_{\theta_t}$ processes the incoming batch $D_t$ and updates its parameters from $\theta_t$ to $\theta_{t+1}$ using an unsupervised proxy loss $\psi$, without access to the ground-truth labels or the supervised task loss $\ell$. The objective of TTA problem is to adjust the model parameters in response to distribution shifts within $\mathcal{S}$ in a timely and reliable manner, thereby improving the performance on the entire test data stream.

As discussed in previous TTA studies (Wang et al., 2021), the effectiveness of TTA relies heavily on the alignment between the proxy loss $\psi$ and the task loss $\ell$. Therefore, we formalize this requirement by introducing the following assumption on alignment and gradient bias locally.

**Assumption 2.1** (($\alpha, \zeta$)-Alignment between Proxy and Task Losses). There exists a radius $r > 0$ such that within the local neighborhood $\mathcal{N}_r(\theta_1) = \{\theta : \|\theta - \theta_1\| \leq r\}$, the expected proxy loss $\psi_t(\theta) = \mathbb{E}_{x \sim \mathcal{P}_{t,X}}[\psi(\theta; x)]$ and the expected task loss $\ell_t(\theta) = \mathbb{E}_{(x,y) \sim \mathcal{P}_t}[\ell(\theta; x, y)]$ satisfy

$$\langle \nabla \psi_t(\theta), \nabla \ell_t(\theta) \rangle \geq \alpha \|\nabla \ell_t(\theta)\|^2 - \zeta, \qquad (1)$$

for any $\theta \in \mathcal{N}_r(\theta_1)$ and some constants $\alpha > 0$ and $\zeta \geq 0$.

**Assumption 2.2** (Regularity of the Loss). There exists a radius $r > 0$ such that within $\mathcal{N}_r(\theta_1) = \{\theta : \|\theta - \theta_1\| \leq r\}$, the task loss $\ell$ is $L$-smooth and satisfies the Polyak–Łojasiewicz condition with parameter $\mu > 0$, while the proxy loss $\psi$ is $L$-smooth and $\mu$-strongly convex. Both $\ell(\theta; \cdot)$ and $\psi(\theta; \cdot)$ are $L_x$-Lipschitz with respect to the data, and $\nabla \psi(\theta; \cdot)$ is $L_{\nabla \psi}$-Lipschitz with respect to the data, i.e.,

$$\|\nabla \psi(\theta; x) - \nabla \psi(\theta; x')\| \leq L_{\nabla \psi}\|x - x'\|, \qquad (2)$$

for all $\theta \in \mathcal{N}_r(\theta_1)$ and all $x, x'$. The stochastic gradients of both $\ell$ and $\psi$ have variance bounded by $\sigma^2$ and norm bounded by $G$, and the iterates $\{\theta_t\}_{t \geq 1}$ produced by the TTA algorithm under analysis remain in $\mathcal{N}_r(\theta_1)$.

*Remark* 2.3. In Assumption 2.1, $\alpha > 0$ ensures that the proxy gradient is sufficiently aligned with a descent direction for the task loss, while $\zeta$ accounts for imperfections of the proxy loss. This assumption guarantees that the proxy gradients provide a constructive signal for reducing the task loss, which is the foundation for TTA to be feasible. Assumption 2.2 imposes standard local assumptions on the losses and gradients, which make the subsequent recovery and learnability analysis tractable. Both assumptions are

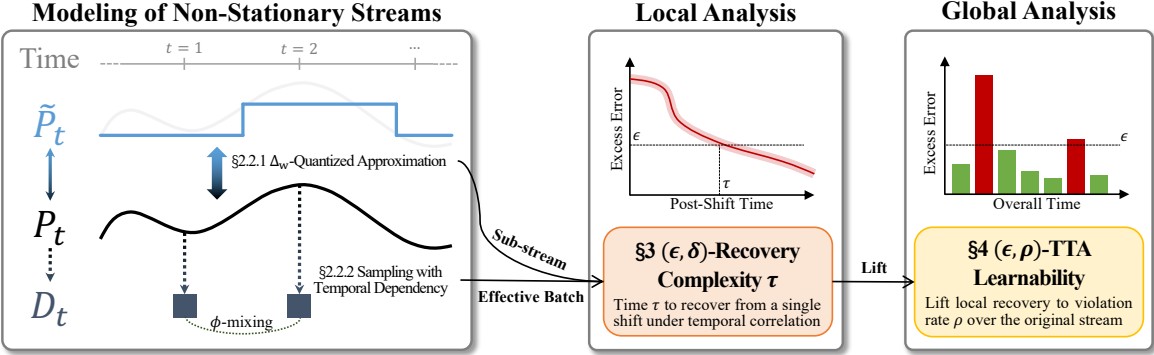

*Figure 1.* Overview of the theoretical modeling and analysis in this paper. A unified model of non-stationary test streams underpins a local analysis via $(\epsilon, \delta)$-Recovery Complexity, which is lifted to a global $(\epsilon, \rho)$-TTA Learnability.

stated on the same neighborhood $\mathcal{N}_r(\theta_1)$, consistent with the local proxy-optimal competitor in Definition 2.11.

However, even with the local Assumptions 2.1 and 2.2 on $\mathcal{N}_r(\theta_1)$, establishing a theoretical foundation for TTA problem remains challenging due to two fundamental issues:

(a) **Limited Modeling of Test Streams**: Data streams in the real world exhibit complex forms of non-stationarity (Zhao et al., 2023). However, existing formalizations of stream dynamics (Garg et al., 2020; Sugiyama et al., 2007; Zhou et al., 2023) either lack sufficient expressiveness to capture the full structure of continuously evolving and temporally dependent streams, or lead to models that are analytically intractable for deriving sharp theoretical guarantees.

(b) **Vagueness in Target Objective**: TTA algorithms adapt to a non-stationary test stream, requiring high accuracy at each time step (Wang et al., 2022). This blind pursuit of performance has led to the neglect of whether the TTA task is well-defined and learnable.

Therefore, to build theoretical foundations for TTA problem, we first formalize the test stream by measuring its underlying distribution shifts in a global view and capturing dependency structure locally to take various types of non-stationarity into account in Section 2.2, and then we define an achievable yet strong competitive target in Section 2.3 for the following theoretical analysis.

## 2.2. Formalization of Test Stream

Test data streams in the real world often encounter complex non-stationarity. On the global view, the environment may change over time, leading to shifts in the underlying data distribution, such as gradually changing lighting or rapidly fluctuating weather conditions (Hendrycks & Dietterich, 2019). Meanwhile, consecutive data points may be highly

correlated on a local view, leading to strong temporal dependencies in the data stream, such as objects appearing in consecutive video frames (Geiger et al., 2013).

Existing TTA studies consider different types of test streams, including target distribution shifts in Tent (Wang et al., 2021), sequential distribution shifts in CoTTA (Wang et al., 2022), temporally correlated non-i.i.d. streams in NOTE (Gong et al., 2022), and open-world data shifts involving both covariate and label distribution shifts in ODS (Zhou et al., 2023). To account for the diversity of test streams, we model the dynamics of the test stream through two complementary dimensions: *Distribution Shift* and *Temporal Correlation*, which respectively capture global changes and local dependencies in the data stream. Specifically, we assume that each batch $D_t$ at time step $t$ is sampled from the underlying data distribution $\mathcal{P}_t$ at time $t$, influenced by the temporal correlation.

### 2.2.1. DISTRIBUTION SHIFT

To unify the analysis of both gradual and abrupt distribution shifts, and to enable principled information-theoretic lower bounds, we adopt the Wasserstein-1 distance (Panaretos & Zemel, 2019) on the joint space $\mathcal{X} \times \mathcal{Y}$ to discretize the distributional trajectory $\{\mathcal{P}_t\}_{t=1}^T$ into a bounded piecewise-constant surrogate that captures continuous and complex distribution shifts.

**Assumption 2.4** ($\Delta_W$-Quantized Distribution Shift). The underlying data distributions $\{\mathcal{P}_t\}_{t=1}^T$ in the test stream satisfy bounded path variation

$$V_T := \sum_{t=1}^{T-1} W_1(\mathcal{P}_t, \mathcal{P}_{t+1}) < \infty, \tag{3}$$

together with the single-step regularity condition

$$W_1(\mathcal{P}_t, \mathcal{P}_{t+1}) \le \Delta_W/2 \tag{4}$$

**Algorithm 1** Greedy Quantized Approximation Method

**Require:** Distribution $\{\mathcal{P}_t\}_{t=1}^T$, resolution $\Delta_W > 0$
**Ensure:** Approximation $\{\tilde{\mathcal{P}}_t\}_{t=1}^T$, shift indicators $\{\tilde{S}_t\}_{t=2}^T$
 1: Set anchor index $a \leftarrow 1$
 2: Set $\tilde{\mathcal{P}}_1 \leftarrow \mathcal{P}_1$
 3: **for** $t = 2$ to $T$ **do**
 4:   **if** $W_1(\mathcal{P}_t, \mathcal{P}_a) \leq \Delta_W/2$ **then**
 5:     $\tilde{\mathcal{P}}_t \leftarrow \mathcal{P}_a$
 6:     $\tilde{S}_t \leftarrow 0$
 7:   **else**
 8:     $\tilde{S}_t \leftarrow 1$ {Declare a shift}
 9:     $a \leftarrow t$ {Start a new sub-stream}
10:     $\tilde{\mathcal{P}}_t \leftarrow \mathcal{P}_t$
11:   **end if**
12: **end for**

for any $t$, which ensures that the resolution $\Delta_W$ is no smaller than the largest single-step drift in the original stream. Fix a resolution parameter $\Delta_W > 0$. We assume there exists a piecewise-constant approximation $\{\tilde{\mathcal{P}}_t\}_{t=1}^T$ of underlying data distributions and shift indicators $\{\tilde{S}_t\}_{t=2}^T$ such that, for each $t = 2, \ldots, T$,

$$
\begin{cases}
W_1(\tilde{\mathcal{P}}_{t-1}, \tilde{\mathcal{P}}_t) = 0, & \tilde{S}_t = 0, \\
0 \leq W_1(\tilde{\mathcal{P}}_{t-1}, \tilde{\mathcal{P}}_t) \leq \Delta_W, & \tilde{S}_t = 1,
\end{cases}
\tag{5}
$$

and the approximation error is uniformly bounded:

$$
\sup_{t \in [T]} W_1(\mathcal{P}_t, \tilde{\mathcal{P}}_t) \leq \Delta_W/2.
\tag{6}
$$

where $W_1(\cdot, \cdot)$ measures the Wasserstein-1 distance.

**Proposition 2.5** (Greedy Approximation and Shift-Count Bound). *Under Assumption 2.4, for any $\Delta_W > 0$, Algorithm 1 produces a piecewise-constant approximation $\{\tilde{\mathcal{P}}_t\}_{t=1}^T$ that satisfies the uniform discrepancy bound Equation (6). Moreover, letting $\tilde{K}_S(T) := \sum_{t=2}^T \mathbb{I}[\tilde{S}_t = 1]$ be the number of shifts, we have the shift-count upper bound*

$$
\tilde{K}_S(T) \leq \left\lceil \frac{2V_T}{\Delta_W} \right\rceil.
\tag{7}
$$

*Remark* 2.6. Proposition 2.5 shows that Algorithm 1 always produces a piecewise-constant approximation valid under Assumption 2.4, and links the total path variation $V_T$ of the original stream to the shift count of the approximation, allowing us to quantify the discrepancy between the stream and its approximation.

### 2.2.2. TEMPORAL CORRELATION

The temporal correlation in the test data stream (Gong et al., 2022) is difficult to formalize as it may arise from diverse and unknown dependency structures across time. Rather than directly modeling such correlations, we adopt a tractable surrogate and measure the strength of temporal dependence by assuming that the test stream is generated by a stochastic process satisfying the $\phi$-mixing condition (Bradley, 1986) with a mixing coefficient $\phi(i)$.

**Assumption 2.7** ($\phi$-Mixing Temporal Dependency). The test stream $\{D_t\}_{t=1}^T$ is generated by a stochastic process whose temporal dependence is controlled by uniformly decaying $\phi$-mixing coefficients. Let $\mathcal{H}_t = \{D_1, \ldots, D_t\}$ be the past history up to time $t$, and let $\mathcal{H}_{t+i}^T = \{D_{t+i}, \ldots, D_T\}$ be the future from time $t + i$ to $T$. Define the $\phi$-mixing coefficient

$$
\phi(i) = \sup_t \sup_{\substack{P \in \sigma(\mathcal{H}_t), \\ Q \in \sigma(\mathcal{H}_{t+i}^T),\, \mathbb{P}(P)>0}} \big|\mathbb{P}(Q \mid P) - \mathbb{P}(Q)\big|. \tag{8}
$$

Moreover, we assume a geometric decay condition of $\phi(i)$, i.e., there exists $\varrho \in [0, 1)$ such that

$$
\phi(i) \leq \varrho^i, \forall i \geq 1.
\tag{9}
$$

Here, $\varrho \in [0, 1)$ is the geometric decay parameter of the mixing coefficients.

*Remark* 2.8. Assumption 2.7 characterizes temporal dependence through mixing coefficients. Smaller $\phi(i)$ indicates weaker temporal correlations between events separated by lag $i$. The geometric decay condition Equation (9) is a standard strengthening that enables closed-form constants depending only on $\varrho$ for simplicity.

With Assumption 2.7, the following Proposition 2.9 shows how temporal correlation affects one of the key factors of TTA, i.e., the batch size $B$.

**Proposition 2.9** (Effective Batch Size Reduction). *Consider a batch of samples collected from a single time window of a $\phi$-mixing process satisfying Assumption 2.7. We define $B_{eff}$ as the size of an i.i.d. batch whose batch-mean gradient matches the resulting variance upper bound $\sigma^2/B_{eff}$ of the temporally correlated batch of size $B$. Then,*

$$
B_{eff} := \frac{B}{C_\phi} \leq B,
\tag{10}
$$

*where $C_\phi := 1 + \dfrac{4\,\varrho^{1/2}}{1 - \varrho^{1/2}}$ and $\varrho$ is the geometric decay parameter from Equation (9).*

*Remark* 2.10. This proposition shows that temporal correlation affects the batch size in the gradient-based model update. Specifically, when batches in $\mathcal{S}$ are temporally independent, then we have $C_\phi = 1$ and $B_{eff} = B$. Otherwise, as temporal correlation increases $C_\phi > 1$, the effective batch size $B_{eff}$ reduces. This highlights the key challenge in TTA with strong temporal correlation, i.e., the effective batch

size is significantly reduced.

## 2.3. Formalization of Competitive Target

In prior TTA studies (Wang et al., 2021; Zhou et al., 2023), algorithms are typically evaluated by their absolute risk on the test stream, implicitly assuming that perfect adaptation is achievable. However, in such an unsupervised problem (Jia et al., 2026a), the proxy loss and task loss are generally misaligned, which fundamentally limits the attainable performance.

To make the TTA objective well-defined, we introduce a proxy-optimal competitor and its associated competitive target risk, which provide a realistic oracle.

**Definition 2.11** (Proxy-Optimal Competitor and Competitive Target). For each time step $t$, define the proxy-optimal parameter within a local neighborhood of the initial parameters $\theta_1$ as

$$\theta_t^\star \in \arg \min_{\theta \in \mathcal{N}_r(\theta_1)} \psi_t(\theta), \qquad (11)$$

where $\mathcal{N}_r(\theta_1) = \{\theta \in \Theta : \|\theta - \theta_1\| \leq r\}$ denotes an $r$-radius neighborhood of $\theta_1$ and $\psi_t(\theta) = \mathbb{E}_{x \sim \mathcal{P}_{t,X}}[\psi(\theta; x)]$, with $r$ chosen so that $\theta_t^\star$ is interior. The corresponding competitive target risk is defined as

$$R_t := \ell_t(\theta_t^\star). \qquad (12)$$

where $\ell_t(\theta) = \mathbb{E}_{(x,y) \sim \mathcal{P}_t}[\ell(\theta; x, y)]$.

*Remark* 2.12. This local competitor reflects the best proxy-optimal model reachable within an $r$-radius neighborhood of the initialization $\theta_1$, reflecting the constrained adaptation of practical TTA algorithms (e.g., Tent (Wang et al., 2021) updates only batch-normalization statistics, and CoTTA (Wang et al., 2022) employs parameter restoration). This is also consistent with Assumptions 2.1 and 2.2, which are unlikely to hold over the entire parameter space. Moreover, the competitive target is evaluated using the supervised task loss, providing a principled upper bound on the achievable performance of TTA problem.

## 3. $(\epsilon, \delta)$-Recovery Complexity

As specified in Assumption 2.4, we adopt a unified modeling framework for a broad class of distribution shifts in the test stream $\mathcal{S}$. This framework enables us to analyze a non-stationary test stream in a piecewise-stationary manner, by studying the constructed approximation $\{\tilde{\mathcal{P}}_t\}_{t=1}^T$ instead of the underlying data-generating distributions $\{\mathcal{P}_t\}_{t=1}^T$.

With the help of this framework, we first investigate the theoretical properties of TTA under a single stationary sub-stream induced by a distribution shift in this section, and subsequently extend the analysis to the entire non-stationary

stream in Section 4.

Specifically, we introduce the notion of $(\epsilon, \delta)$-recovery complexity to measure the minimum number of time steps required for a TTA algorithm to reduce its excess risk below $\epsilon$ with probability at least $1 - \delta$ after a distribution shift, assuming the post-shift distribution is stationary.

For simplicity, we focus on a single distribution shift occurring at time $t = 1$, where the TTA algorithm is initialized with a parameter $\theta_1$ well optimized under some pre-shift distribution. The following definition formalizes the $(\epsilon, \delta)$-recovery complexity.

**Definition 3.1** ($(\epsilon, \delta)$-Recovery Complexity). Consider a test-time adaptation problem in which an algorithm $\mathcal{A}$ starts from an initial parameter $\theta_1$. A distribution shift occurs at time $t = 1$, and the distribution remains stationary thereafter.

For any target excess error $\epsilon > 0$ and a constant failure probability budget $\delta$, the $(\epsilon, \delta)$-*Recovery Complexity* of algorithm $\mathcal{A}$, denoted by $\tau(\epsilon, \delta)$, is defined as

$$\tau(\epsilon, \delta) := \inf\left\{ t \in \mathbb{N} : \sup_{u \geq t} \mathbb{P}(\mathcal{E}_u > \epsilon) \leq \delta \right\}, \quad (13)$$

where $\mathcal{E}_t := \ell_t(\theta_t) - R_t$ denotes the excess risk at time $t$. That is, $\tau$ is the smallest time after which the marginal failure probability stays uniformly $\leq \delta$.

*Remark* 3.2. The $(\epsilon, \delta)$-recovery complexity is a meaningful performance metric for test-time adaptation problems for two main reasons. First, unlike standard online learning settings (Shalev-Shwartz, 2012) that focus on average performance, TTA requires the model to maintain satisfactory performance throughout the test stream. An algorithm that achieves low average error but exhibits long periods of poor performance is undesirable in practice. The recovery complexity explicitly quantifies the duration during which the model's excess risk exceeds a pre-defined threshold, thereby measuring and controlling the length of unsatisfactory periods in the test stream $\mathcal{S}$. Second, in realistic test-time scenarios, the data stream is generally non-stationary and may undergo multiple distribution shifts. Since long-term stationarity cannot be assumed, recovery-based metrics, such as $(\epsilon, \delta)$-recovery complexity, provide a more appropriate metric for evaluating the effectiveness of TTA algorithms.

### 3.1. Problem-Level Minimax Lower Bound

In this section, we explore the fundamental limits of test-time adaptation problem. Specifically, we investigate the conditions under which recovery from a distribution shift is possible for any TTA algorithm operating in the stochastic proxy-gradient oracle model, and we derive the corresponding minimax lower bounds on recovery complexity. Here,

the oracle model provides only unlabeled batch-level proxy-gradient estimates, not labels or task-loss values.

To this end, we adopt the classical two-point testing method (Le Cam, 2012) to derive minimax lower bounds on the $(\epsilon, \delta)$-recovery complexity from an information-theoretic perspective. The following theorem delineates the regimes in which recovery from distribution shift is provably impossible, and illustrates how key factors govern the intrinsic difficulty of recovery in test-time adaptation.

**Theorem 3.3** (Minimax Lower Bound on $(\epsilon, \delta)$-Recovery Complexity). *Suppose that Assumptions 2.1, 2.2, 2.4, and 2.7 hold for the test-time adaptation problem, all stated locally over $\mathcal{N}_r(\theta_1)$.*

*For any target excess error $\epsilon > 0$, failure probability budget $\delta \in (0, 0.5)$, and shift budget $\Delta_W \geq 2\sqrt{\zeta/\alpha + 2\epsilon} + 2\sqrt{\zeta/\alpha}$, the $(\epsilon, \delta)$-Recovery Complexity $\tau(\epsilon, \delta)$ for any TTA algorithm operating in the stochastic proxy-gradient oracle model must satisfy*

$$\tau \geq \Omega\left(\frac{C_\phi}{B} \cdot \frac{1}{\alpha\left(\sqrt{\zeta + 2\alpha\epsilon} + \sqrt{\zeta}\right)^2}\right) \quad (14)$$

*where the hidden constant may depend on $\delta$ and $\sigma$.*

*Remark* 3.4 (Error Floor raised by $\zeta$). The lower bound exhibits an explicit dependence on the target excess error $\epsilon$ and the misalignment term $\zeta$ through the term $\alpha\left(\sqrt{\zeta + 2\alpha\epsilon} + \sqrt{\zeta}\right)^2$. When $\zeta = 0$, corresponding to a perfectly aligned proxy loss, this term reduces to $\Theta(\alpha^2\epsilon)$, yielding $\tau \geq \Omega\left(\frac{C_\phi}{B\alpha^2\epsilon}\right)$, which matches the order of standard stochastic optimization lower bounds (Agarwal et al., 2012). In contrast, when $\zeta > 0$, the lower bound retains a non-vanishing misalignment-induced component as $\epsilon \to 0$, reflecting an error floor induced by an imperfect proxy. This highlights that TTA is constrained by the misalignment.

*Remark* 3.5 (Joint Effects of $B$ and $C_\phi$). The linear dependence on the inverse batch size $1/B$, highlighting that the batch size is a key factor in TTA problem. Moreover, the appearance of the $C_\phi$ indicates that temporal correlation in the test stream effectively reduces informative samples in each batch. This shows that TTA is constrained jointly by batch size and temporal correlation.

*Remark* 3.6 (Quadratic Dependence on $\alpha$). The alignment strength $\alpha$ controls how strongly the proxy gradient reflects a descent signal for the task loss. Since the information carried by noisy proxy-gradient observations scales quadratically with this signal strength, the recovery complexity scales as $1/\alpha^2$ in the well-aligned case. Together with the matching upper bound in Theorem 3.9, stronger proxy-task alignment thus quadratically reduces the recovery time.

*Remark* 3.7 (No Explicit Dependence on $\Delta_W$). The minimax lower bound does not explicitly depend on the distribu-

tion shift magnitude $\Delta_W$. This is because the lower bound is established via a constructed hard instance that already satisfies the assumed constraint on $\Delta_W$. Once the distribution shift is sufficient to move the model away from the current optimum, the intrinsic difficulty of TTA is governed by the above-discussed factors rather than the magnitude of the shift. Our Theorem 3.3 is non-vacuous only when the distribution shift is large enough that adaptation is required to achieve the target excess error $\epsilon$. If $\Delta_W$ is sufficiently small, the target accuracy may be satisfied without adaptation, in which case any lower bound becomes vacuous.

### 3.2. Algorithm-Level Upper Bound

In the previous section, we established a minimax lower bound on the $(\epsilon, \delta)$-recovery complexity for any TTA algorithm in the stochastic proxy-gradient oracle model in Theorem 3.3, illustrating the fundamental difficulty of TTA. To complement this problem-level lower bound and assess its tightness, we analyze a simple TTA baseline.

Specifically, we derive an upper bound on the $(\epsilon, \delta)$-recovery complexity for a simple yet representative TTA baseline in Definition 3.8. The theoretical results in Theorem 3.9 serve two purposes: (a) it quantifies the achievable recovery complexity of practical algorithms; (b) it reveals the remaining gap between the minimax lower bound and actual algorithmic performance, thereby highlighting potential directions for future TTA studies.

**Definition 3.8** (TTA Baseline). The *TTA baseline* operates on a test data stream of batches $\{D_t\}_{t=1}^T$ as follows. At each time step $t$, given the current model parameters $\theta_t$, the algorithm: (1) provides predictions on the batch $D_t$ based on $\theta_t$; (2) updates the parameters $\theta_t$ via a single stochastic gradient step on the proxy loss $\theta_{t+1} \leftarrow \theta_t - \eta\nabla\hat{\psi}_t(\theta_t)$, where $\nabla\hat{\psi}_t(\theta_t)$ denotes a stochastic mini-batch estimator of $\nabla\psi_t(\theta_t)$ computed from $D_t$, and $\eta > 0$ is a fixed learning rate. As is standard in stochastic-gradient analyses, we treat $\nabla\hat{\psi}_t(\theta_t)$ as conditionally unbiased given the past, with effective variance bounded as in Proposition 2.9.

**Theorem 3.9** (Upper Bound on $(\epsilon, \delta)$-Recovery Complexity). *Suppose that Assumptions 2.1, 2.2, 2.4, and 2.7 hold for the test-time adaptation problem, all stated locally over $\mathcal{N}_r(\theta_1)$. The initial excess error of $\theta_1$ on the pre-shift distribution $\mathcal{P}^-$ is bounded by $\epsilon$.*

*For any target excess error $\epsilon > 0$ and a constant failure probability budget $\delta \in (0, 0.5)$, if the alignment bias satisfies $\frac{\zeta}{\alpha\mu} \leq \frac{\epsilon\delta}{2}$ and the canonical regime $BG^2 \leq \sigma^2 C_\phi$ holds, then the $(\epsilon, \delta)$-Recovery Complexity $\tau(\epsilon, \delta)$ for the TTA baseline in Definition 3.8 with an appropriately chosen learning rate $\eta$ satisfies*

$$\tau \leq \mathcal{O}\left(\frac{C_\phi}{B\alpha^2\epsilon}\log\left(\frac{\Delta_W + \epsilon}{\epsilon}\right)\right). \qquad (15)$$

*Here, the constants $L$, $\mu$, $L_x$, $\sigma$, and the failure probability budget $\delta$ are treated as problem-dependent constants and are omitted in the Big-O notation. The precise dependence on $\delta$ is detailed in Equation (78) of the proof.*

*Remark* 3.10 (Feasibility Condition for $(\epsilon, \delta)$-Recovery). The condition $\zeta/(\alpha\,\mu) \leq \epsilon\,\delta/2$ reveals a minimal feasibility condition for high-probability recovery of TTA problem. It ensures that the systematic bias induced by misalignment of proxy loss is dominated by the target accuracy and probability requirements. When this condition is violated, the proxy loss fails to provide reliable guidance toward the true objective, and no algorithm can guarantee $(\epsilon, \delta)$-recovery regardless of the recovery time.

*Remark* 3.11 (Logarithmic Dependence on $\Delta_W$). The logarithmic factor $\log((\Delta_W + \epsilon)/\epsilon)$ arises from eliminating the initial excess error induced by the distribution shift. Under smoothness assumptions, the error decreases geometrically, requiring a logarithmic number of iterations to reduce the error from scale $\Delta_W$ to the target accuracy $\epsilon$.

*Remark* 3.12 (Near-Minimax Optimality of Baseline). Up to the logarithmic factor, the upper bound matches the minimax lower bound in its dependence on the alignment strength $\alpha$, target excess error $\epsilon$, and batch size $B$, indicating that the baseline achieves the optimal recovery complexity. The remaining logarithmic gap arises from initialization effects rather than suboptimal adaptation dynamics. This sharp matching holds in the canonical TTA regime characterized by non-negligible temporal correlation, formally $C_\phi/B \geq G^2/\sigma^2$, which is the parameter range targeted by our analysis. In the i.i.d.-like opposite limit, where $C_\phi$ approaches its minimum and $B$ is large, the proxy-gradient norm contributes an additive constant warm-up factor that we absorb into the Big-O notation; this regime corresponds to the classical i.i.d. setting and is not the focus of this work.

To summarize, the matching upper and lower bounds characterize the fundamental limits of test-time adaptation using proxy loss and gradient updates under the considered assumptions, **leaving little room for algorithmic improvement for test-time adaptation without strengthening the feedback or adding structural assumptions.**

## 4. Learnability of Test-Time Adaptation

In the previous section, we have established the $(\epsilon, \delta)$-recovery complexity of test-time adaptation by analyzing the minimal recovery time within each stationary sub-stream induced by a distribution shift, under our approximation framework in Assumption 2.4. These results provide a precise understanding of how TTA algorithms can recover to a target accuracy after an individual distribution shift.

Building upon the local recovery analysis, we now study the global behavior of test-time adaptation algorithms over the entire test stream. To this end, we introduce a novel $(\epsilon, \rho)$-*TTA learnability* framework that formalizes long-term reliability for TTA problem. Our analysis connects piecewise recovery guarantees to cumulative performance over time and quantifies the effects introduced by the approximation framework used to model distributional shifts. In this way, recovery complexity is lifted to a principled notion of global learnability. Finally, we relate our learnability framework to dynamic regret, clarifying both the connections and the fundamental distinctions between the TTA problem and standard learning problem settings.

**Definition 4.1** (($\epsilon, \rho$)-TTA Learnability)**.** Consider a test-time adaptation problem defined over a test stream with underlying data distributions $\{\mathcal{P}_t\}_{t=1}^T$. For a target excess error $\epsilon > 0$ and a tolerance parameter $\rho \in (0, 1)$, the problem is said to be $(\epsilon, \rho)$-learnable if there exists a TTA algorithm such that

$$\frac{1}{T}\sum_{t=1}^T \mathbb{P}(\ell_t(\theta_t) - R_t > \epsilon) \leq \rho, \qquad (16)$$

where the probability is taken with respect to the algorithmic randomness and data sampling, and $\rho$ represents the maximum allowable fraction of time steps at which the target excess error $\epsilon$ is violated.

*Remark* 4.2. The notion of $(\epsilon, \rho)$-TTA learnability quantifies the long-term reliability of TTA algorithms by controlling the fraction of time steps at which the target excess error $\epsilon$ is exceeded. Specifically, it ensures that the algorithm maintains acceptable performance for all but a $\rho$ fraction of the test stream. This formulation aligns closely with practical deployment scenarios, such as autonomous driving (Gong et al., 2022) and financial trading (Zhou et al., 2025a), where TTA algorithms operate in dynamical environments and must ensure low risk for the majority time.

### 4.1. From Recovery Complexity to Learnability

In this section, we aim to bridge the gap between the local notion of $(\epsilon, \delta)$-recovery complexity in Definition 3.1 and the global notion of $(\epsilon, \rho)$-TTA learnability in Definition 4.1.

While recovery complexity provides an accurate bound on the adaptation time within a stationary sub-stream, extending these guarantees to the entire test stream presents the following two nontrivial challenges: (a) The theoretical results on recovery complexity are built on a quantized approximation, assuming that the distribution shift is discrete, which may not hold for real-world data streams. How to connect the quantized approximation with the original stream? (b)

The quantized approximation inevitably introduces a mismatch between the approximated and true data distribution. How to quantify this discrepancy?

To address these challenges, we first connect the original and quantized streams via Proposition 2.5, which establishes a relationship between the path variation $V_T$ of the original stream and the number of shifts in the quantized stream. We then derive a pointwise discrepancy between the original and the quantized stream at each time step to accurately quantify the approximation error. Finally, we present Theorem 4.3, which converts the $(\epsilon, \delta)$-recovery complexity upper bound of a TTA algorithm on the quantized stream into a learnability for this algorithm on the original stream.

**Theorem 4.3** (From Recovery to Learnability). *Suppose Assumptions 2.1, 2.2, 2.4, and 2.7 hold, and define the bridging constant $\Lambda := L_x + GL_{\nabla\psi}/\mu$. Fix a target excess error $\epsilon > \Lambda\Delta_W$ and a failure probability budget $\delta$, and set $\epsilon' := \epsilon - \Lambda\Delta_W$. Let $\{\theta_t\}_{t=1}^{T}$ be the iterates of the TTA algorithm on the test stream $\{\mathcal{P}_t\}_{t=1}^{T}$, and suppose the algorithm attains an $(\epsilon', \delta)$-recovery complexity $\tau(\epsilon', \delta)$ on each stationary sub-stream of the quantized approximation. Then the algorithm is $(\epsilon, \rho)$-learnable with respect to the competitive target $R_t = \ell_t(\theta_t^\star)$, where*

$$\rho \leq \delta + \frac{(\tilde{K}_S(T) + 1)\,\tau(\epsilon', \delta)}{T}, \qquad (17)$$

*and $\tilde{K}_S(T)$ denotes the number of shifts in Equation (7).*

*Remark* 4.4 (Feasibility Conditions). The learnability guarantee in Theorem 4.3 relies on two complementary feasibility conditions. First, the condition $\epsilon > \Lambda\Delta_W$ with $\Lambda = L_x + GL_{\nabla\psi}/\mu$ ensures that the target excess error on the original stream is achievable under the approximation framework, accounting for the discrepancy between the original and quantized streams. Second, the alignment condition $\frac{\zeta}{\alpha\mu} \leq \frac{(\epsilon - \Lambda\Delta_W)\delta}{2}$ ensures that the systematic error caused by misalignment between the proxy loss and target loss does not dominate the recovery process. When this condition is violated, the proxy loss becomes insufficiently informative to guarantee recovery to accuracy $\epsilon$ with probability at least $1 - \delta$, even on the quantized stream.

*Remark* 4.5 (Strong Transfer from Recovery to Learnability). Theorem 4.3 establishes a strong transfer principle from local recovery guarantees to global learnability. Specifically, any improvement in recovery complexity, such as faster recovery or fewer shifts in the quantized stream, directly translates into a smaller violation rate $\rho$. This result shows that recovery complexity is a fundamental building block governing the long-term reliability of TTA. This result also provides a principled way to design TTA algorithms with global learnability guarantees by focusing on improving local recovery complexity, which is more tractable to analyze.

## 4.2. From Learnability to Dynamic Regret

Having established a connection between $(\epsilon, \delta)$-recovery complexity and $(\epsilon, \rho)$-learnability, we have shown that the long-term reliability of a TTA algorithm can be guaranteed by its local recovery complexity on each stationary sub-stream of a quantized approximation.

We now further connect our learnability framework with existing learning paradigms by relating it to dynamic regret, a central notion in online learning for non-stationary environments (Shalev-Shwartz, 2012; Zhang et al., 2018b; Zhao et al., 2024). This result clarifies both the connections and the fundamental differences between test-time adaptation problem and online learning problem.

**Theorem 4.6** (From Learnability to Dynamic Regret). *Assume that the excess risk is uniformly bounded. There exists a constant $M > 0$ such that*

$$0 \leq \ell_t(\theta_t) - R_t \leq M \quad \text{almost surely for all } t. \quad (18)$$

*If the problem is $(\epsilon, \rho)$-learnable in Definition 4.1, that is,*

$$\frac{1}{T}\sum_{t=1}^{T}\mathbb{P}(\ell_t(\theta_t) - R_t > \epsilon) \leq \rho, \qquad (19)$$

*then the expected cumulative dynamic regret satisfies*

$$\text{Reg}(T) := \sum_{t=1}^{T}\mathbb{E}[\ell_t(\theta_t) - R_t] \leq T(\epsilon + M\rho), \quad (20)$$

*where $R_t$ denotes the comparator in Definition 2.11.*

*Remark* 4.7 (Difference #1 from Online Learning). Theorem 4.3 requires $\epsilon > \Lambda\Delta_W$, where $\Lambda = L_x + GL_{\nabla\psi}/\mu$. Thus, when the approximation discrepancy does not vanish, i.e., $\Delta_W = \Omega(1)$, the theorem only certifies recovery at a constant excess-risk level. Consequently, the regret conversion in Theorem 4.6 yields at best a linear bound $\text{Reg}(T) = \mathcal{O}(T)$ through this framework. This reflects a key distinction from standard dynamic-regret analyses, which typically obtain sublinear guarantees only under additional regularity of the comparator or environment variation.

*Remark* 4.8 (Difference #2 from Online Learning). TTA also differs from online learning in its supervision. The learner observes unlabeled data and updates through a proxy loss rather than the task loss itself. The alignment parameters $\alpha$ and $\zeta$ in Assumption 2.1 quantify this information constraint. In particular, imperfect alignment introduces a systematic bias that cannot be removed by increasing the recovery time alone. When shifts occur frequently, e.g., $\tilde{K}_S(T) = \mathcal{O}(T)$, the accumulated recovery cost $\tilde{K}_S(T)\tau(\epsilon', \delta)$ therefore reflects an intrinsic cost of the unsupervised adaptation.

*Remark* 4.9 (Connection to Online Learning). The proposed

framework recovers an online-learning-like regime when recovery effects are negligible. If the stream has vanishing shifts and a bounded recovery time, i.e., $\Delta_W \to 0$ and $\tilde{K}_S(T) = o(T)$ with $\tau(\epsilon', \delta) = \mathcal{O}(1)$, then Theorem 4.3 gives $\rho = o(1)$ for suitable $\delta = o(1)$. With Theorem 4.6,

$$\text{Reg}(T) \leq T(\epsilon + M\rho),$$

so choosing $\epsilon = o(1)$ yields $\text{Reg}(T) = o(T)$. Thus, our framework connects to classical dynamic-regret analysis in benign regimes, while exposing additional recovery and proxy-supervision costs in TTA.

## 5. Empirical Validation

We empirically validate our theoretical results in both controlled synthetic environments and real-world benchmarks. We summarize the main conclusions below and defer the experimental setup and detailed results to Appendix G.

### 5.1. Controlled Synthetic Environments

We conduct controlled experiments on a one-dimensional recovery problem, with the setup detailed in Appendix G.1. The results in Appendix G.2 show that the measured recovery time $\tau$ follows the predicted scaling $\tau = \mathcal{O}(1/\alpha^2)$ and $\tau = \mathcal{O}(1/B)$, and stays above the minimax lower-bound in Theorem 3.3 and follows the same scaling as the algorithmic upper-bound in Theorem 3.9, supporting the near-tightness.

### 5.2. Real-world Benchmarks

We also conduct experiments on the real-world benchmarks CIFAR-10-C, CIFAR-100-C, and ImageNet-C, with the setup detailed in Appendix G.3. As reported in Appendix G.4, the empirical proxy-task alignment $\tilde{\alpha}$ is positive on every benchmark, and Tent (Wang et al., 2021) accordingly improves accuracy on all 15 corruptions of each dataset, confirming that the alignment assumption (Assumption 2.1) underlying our analysis holds for a standard TTA algorithm. Conversely, under strong temporal correlation the empirical alignment turns negative and Tent collapses, providing empirical evidence that the alignment is the core factor governing whether TTA succeeds in practice.

## 6. Related Work

### 6.1. Test-Time Adaptation

Test-time adaptation (Liang et al., 2025; Wang et al., 2025; Xiao & Snoek, 2024) aims to adapt a source model to distribution shifts in test data without accessing source data. Classical TTA studies optimize prediction entropy (Wang et al., 2021), and following approaches (Niu et al., 2022; Yuan et al., 2023; Wang et al., 2022) improve adaptation stability. Another line of methods studies TTA in the open world, such

as LAME (Boudiaf et al., 2022), NOTE (Gong et al., 2022), ODS (Zhou et al., 2023), SAR (Niu et al., 2023), etc. Existing studies also extend TTA to MLLMs (Jia et al., 2026b; You et al., 2026) and LLMs (Shao et al., 2026; Zhou et al., 2025b), such as AdaPrompt (Zhang et al., 2024a), TPT (Shu et al., 2022), TLM (Hu et al., 2025), etc. Theoretical analyses on TTA include gradient-correlation and task-relation guarantees (Sun et al., 2020; Liu et al., 2021), error bounds on memory banks (Zhang et al., 2023), active sample selection and label queries (Gui et al., 2024), dynamic regret in non-stationary environments (Zhang et al., 2024b), provable collapse on Gaussian mixtures (Hoang et al., 2024), and anytime-valid shift-detection guarantees (Bar et al., 2024). However, these studies cannot provide a global learnability guarantee that jointly handles unlabeled supervision, non-stationary streams, and temporal correlation, making them less applicable in the real world.

### 6.2. Online Learning

Online learning in non-stationary environments controls dynamic regret against time-varying comparators (Zinkevich, 2003; Zhang et al., 2025), with subsequent work extending to path-length–adaptive (Zhang et al., 2018a), strongly adaptive and parameter-free (Hazan & Seshadhri, 2009; Foster et al., 2017), and problem-dependent bounds (Zhao et al., 2020; 2024) under convex losses. However, dynamic regret controls only average performance, not the per-step satisfaction targeted by our recovery complexity and learnability.

## 7. Conclusion

We introduced the first learnability framework for test-time adaptation under unlabeled non-stationary test streams, modeling global shifts via a quantized Wasserstein approximation and local dependence via a $\phi$-mixing process. Within this framework, we defined $(\epsilon, \delta)$-recovery complexity and derived nearly matching minimax lower and upper bounds, revealing how proxy alignment, target accuracy, and batch size govern recovery. Building on this, we introduced $(\epsilon, \rho)$-TTA learnability and connected it to dynamic regret, showing that persistent shifts force linear regret and that unlabeled supervision incurs an intrinsic cumulative cost. Empirical results on synthetic and real-world benchmarks confirm our predictions. We hope our framework and primary results will provide a principled foundation for future theoretical and algorithmic developments in test-time adaptation.

**Future work and Limitations.** This work focuses on establishing a general learnability framework for test-time adaptation under unlabeled non-stationary test streams. Our analysis relies on several structural assumptions, which may not fully capture all real-world scenarios. While Section 5 validates our main predictions, a comprehensive empirical study and large-scale benchmarks are left for future work.

## Acknowledgements

This research was supported by the Jiangsu Science Foundation (BK20243012, BG2024036, BK20232003), the National Natural Science Foundation of China (Grant No.624B2068, 62576162), the Fundamental Research Funds for the Central Universities (022114380023), and the "111 Center" (No.B26023).

## Impact Statement

This paper develops a theoretical framework for analyzing the learnability of test-time adaptation. Our results clarify when test-time adaptation is feasible and quantify the intrinsic limitations imposed by distribution shifts, temporal dependence, and imperfect proxy supervision. By introducing the notions of recovery complexity and learnability, this work provides conceptual guidance for the principled design and evaluation of future test-time adaptation algorithms. As the contributions are primarily theoretical and do not introduce new algorithms or systems, we do not anticipate direct negative societal impacts.

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

**APPENDIX CONTENTS**

## A. Proof of Proposition 2.5

*Proof.* Within any sub-stream anchored at time $a_k$, the algorithm sets $\tilde{\mathcal{P}}_t = \mathcal{P}_{a_k}$ for all times $t$ in that sub-stream. The sub-stream is maintained exactly while $W_1(\mathcal{P}_t, \mathcal{P}_{a_k}) \leq \Delta_W/2$. Therefore, for all $t$,

$$W_1(\mathcal{P}_t, \tilde{\mathcal{P}}_t) = W_1(\mathcal{P}_t, \mathcal{P}_{a_k}) \leq \Delta_W/2, \tag{21}$$

which proves Equation (6). Moreover, at any shift time $t_i$, denoting the previous anchor by $t_{i-1}$ (with $t_0 := 1$ for the first sub-stream), the previous-step bound combined with the single-step regularity Equation (4) gives

$$W_1(\tilde{\mathcal{P}}_{t_i-1}, \tilde{\mathcal{P}}_{t_i}) = W_1(\mathcal{P}_{t_{i-1}}, \mathcal{P}_{t_i}) \leq W_1(\mathcal{P}_{t_{i-1}}, \mathcal{P}_{t_i-1}) + W_1(\mathcal{P}_{t_{i-1}}, \mathcal{P}_{t_i}) \leq \tfrac{\Delta_W}{2} + \tfrac{\Delta_W}{2} = \Delta_W, \tag{22}$$

matching the shift-magnitude bound in Assumption 2.4.

Let the shift times be $2 \leq t_1 < t_2 < \cdots < t_m \leq T$, i.e., $\tilde{S}_{t_i} = 1$ for each $i \in [m]$, and thus $\tilde{K}_S(T) = m$. Define also $t_0 := 1$. By the greedy rule, right before the shift at time $t_i$, the current anchor equals $t_{i-1}$, and the shift is triggered because

$$W_1(\mathcal{P}_{t_i}, \mathcal{P}_{t_{(i-1)}}) > \Delta_W/2. \tag{23}$$

On the other hand, by the triangle inequality along the original path,

$$W_1(\mathcal{P}_{t_{(i-1)}}, \mathcal{P}_{t_i}) \leq \sum_{t=t_{(i-1)}}^{t_i-1} W_1(\mathcal{P}_t, \mathcal{P}_{t+1}). \tag{24}$$

Combining (23) and (24) yields, for each $i \in [m]$,

$$\sum_{t=t_{i-1}}^{t_i-1} W_1(\mathcal{P}_t, \mathcal{P}_{t+1}) > \Delta_W/2. \tag{25}$$

The intervals $[t_{i-1}, t_i)$ are disjoint and contained in $\{1, 2, \ldots, T-1\}$. Summing (25) over $i = 1, \ldots, m$ gives

$$m \cdot \frac{\Delta_W}{2} < \sum_{i=1}^{m} \sum_{t=t_{i-1}}^{t_i-1} W_1(\mathcal{P}_t, \mathcal{P}_{t+1}) \leq \sum_{t=1}^{T-1} W_1(\mathcal{P}_t, \mathcal{P}_{t+1}) = V_T.$$

Hence $m < 2V_T/\Delta_W$, i.e.,

$$\tilde{K}_S(T) = m \leq \left\lceil \frac{2V_T}{\Delta_W} \right\rceil,$$

which proves (7). $\square$

## B. Proof of Proposition 2.9

We first give the following Lemma B.1 as the key lemma for proving Proposition 2.9.

**Lemma B.1** ($\phi$-*Mixing Covariance Control,* Bradley, 1986). *Let* $\{Z_t\}_{t \in \mathbb{Z}}$ *be a* $\phi$-*mixing process with coefficients* $\phi(\cdot)$. *For any* $t < s$ *and any square-integrable real-valued, centered functions* $f, g$ *(i.e.,* $\mathbb{E}[f(Z_t)] = \mathbb{E}[g(Z_s)] = 0$*), it holds that*

$$\left| \operatorname{Cov}(f(Z_t), g(Z_s)) \right| \leq 2\sqrt{\operatorname{Var}(f(Z_t)) \operatorname{Var}(g(Z_s))} \, \phi(s - t)^{1/2}. \tag{26}$$

Then, we give the formal proof of Proposition 2.9 as follows.

*Proof.* Consider a mini-batch of size $B$ collected from a single time window $\{Z_1, \ldots, Z_B\}$ of a $\phi$-mixing process satisfying Assumption 2.7. Let $\hat{g}_t \in \mathbb{R}^d$ be the per-sample (or per-instance) stochastic gradient, and assume the centered gradient has bounded second moment:

$$\mathbb{E}[\hat{g}_t] = \nabla \psi(\theta), \qquad \mathbb{E}\left[ \|\hat{g}_t - \nabla \psi(\theta)\|_2^2 \right] \leq \sigma^2. \tag{27}$$

Define the batch-mean gradient estimator $\bar{g} := \frac{1}{B} \sum_{t=1}^{B} \hat{g}_t$. Then its mean-squared error satisfies

$$\mathbb{E}\left[ \|\bar{g} - \nabla \psi(\theta)\|_2^2 \right] \leq \frac{\sigma^2}{B} C_\phi, \qquad C_\phi := 1 + \frac{4 \, \varrho^{1/2}}{1 - \varrho^{1/2}}. \tag{28}$$

Let $x_t := \hat{g}_t - \nabla \psi(\theta)$ so that $\mathbb{E}[x_t] = 0$ and $\mathbb{E}\|x_t\|_2^2 \leq \sigma^2$. We bound the second moment of the average:

$$\mathbb{E}\left\| \frac{1}{B} \sum_{t=1}^{B} x_t \right\|_2^2 = \frac{1}{B^2} \sum_{t=1}^{B} \sum_{s=1}^{B} \mathbb{E}\langle x_t, x_s \rangle = \frac{1}{B^2} \left( \sum_{t=1}^{B} \mathbb{E}\|x_t\|_2^2 + 2 \sum_{1 \leq t < s \leq B} \mathbb{E}\langle x_t, x_s \rangle \right). \tag{29}$$

By the uniform variance bound, $\mathbb{E}\|x_t\|_2^2 \leq \sigma^2$ for all $t$. For the cross terms, fix $t < s$ and decompose into coordinates, $\mathbb{E}\langle x_t, x_s \rangle = \sum_{i=1}^{d} \mathbb{E}[x_t^{(i)} x_s^{(i)}]$. Applying Lemma B.1 to each scalar pair $(x_t^{(i)}, x_s^{(i)})$, summing over $i$, and using Cauchy–Schwarz on $\mathbb{R}^d$ together with $\sum_i \operatorname{Var}(x_t^{(i)}) = \mathbb{E}\|x_t\|_2^2$ (since $x_t$ is centered), we obtain

$$\left| \mathbb{E}\langle x_t, x_s \rangle \right| \leq 2\sqrt{\mathbb{E}\|x_t\|_2^2 \, \mathbb{E}\|x_s\|_2^2} \, \phi(s - t)^{1/2} \leq 2\sigma^2 \phi(s - t)^{1/2}. \tag{30}$$

Plugging (30) into (29) gives

$$\mathbb{E}\left\| \frac{1}{B} \sum_{t=1}^{B} x_t \right\|_2^2 \leq \frac{1}{B^2} \left( B\sigma^2 + 4\sigma^2 \sum_{1 \leq t < s \leq B} \phi(s - t)^{1/2} \right).$$

Re-index by lag $\tau = s - t$ and use the geometric decay (9), so that $\phi(\tau)^{1/2} \leq \varrho^{\tau/2}$:

$$\sum_{1 \leq t < s \leq B} \phi(s - t)^{1/2} = \sum_{\tau=1}^{B-1} (B - \tau)\phi(\tau)^{1/2} \leq B \sum_{\tau=1}^{\infty} \varrho^{\tau/2} = B \cdot \frac{\varrho^{1/2}}{1 - \varrho^{1/2}}.$$

Therefore,

$$\mathbb{E}\left\| \frac{1}{B} \sum_{t=1}^{B} x_t \right\|_2^2 \leq \frac{\sigma^2}{B^2} \left( B + 4B \cdot \frac{\varrho^{1/2}}{1 - \varrho^{1/2}} \right) = \frac{\sigma^2}{B} \left( 1 + \frac{4\varrho^{1/2}}{1 - \varrho^{1/2}} \right).$$

Finally, since $\bar{g} - \nabla \psi(\theta) = \frac{1}{B} \sum_{t=1}^{B} x_t$, we obtain (28), and hence $B_{\text{eff}} = B/C_\phi$. $\qquad \square$

## C. Proof of Theorem 3.3

*Proof.* We reduce the problem to a binary hypothesis testing framework and apply Le Cam's method (Le Cam, 2012) to derive a minimax lower bound on the recovery time $\tau$. Specifically,

$$\inf_{\mathcal{A}} \max_{i \in \{0,1\}} \mathbb{P}_{X^{(i)} \sim \mathcal{P}^{(i)}} \left[ L^{(i)}\left( \mathcal{A}(X^{(i)}) \right) > \epsilon \right] \geq \frac{1}{2}\left( 1 - \operatorname{TV}(X^{(0)}, X^{(1)}) \right). \tag{31}$$

Here, $X^{(i)}$ denotes the observation generated under the $i$-th hypothesis with underlying distribution $\mathcal{P}^{(i)}$, and $L^{(i)}$ is the corresponding loss function. We assume the success regions of the two hypotheses are disjoint.

In the following proof, we omit the time index $t$ for simplicity, assuming a shift occurs at time $t$ and considering the stationary distribution thereafter to compute the lower bound for $\tau$.

### C.0.1. CONSTRUCTING THE BINARY HYPOTHESIS TESTING PROBLEM

We construct a hard instance over a simplified 1-D parameter space $\Theta \subseteq \mathbb{R}$ contained in $\mathcal{N}_r(\theta_1)$, on which we then verify that all assumptions of Theorem 3.3 hold; the lower bound therefore holds against *any* algorithm that operates under those assumptions. After a shift, the underlying distribution is either $\mathcal{P}^{(0)}$ or $\mathcal{P}^{(1)}$, and the hypothesis testing problem requires determining which distribution is active.

We first construct the hard instance. We *define* the supervised loss as $\ell(\theta) = \frac{1}{2}(\theta - m)^2$, where $m$ is the location parameter of the underlying distribution (renamed from the conventional $\mu$ to avoid clash with the Polyak–Łojasiewicz constant in Assumption 2.2):

- Under $\mathcal{P}^{(0)}$, $m = 0$;

- Under $\mathcal{P}^{(1)}$, $m = \Delta$, where $\Delta := 2\rho_{\text{sat}}$ with $\rho_{\text{sat}} := \sqrt{\zeta/\alpha + 2\epsilon} + \sqrt{\zeta/\alpha}$.

We *define* the gradient of the proxy loss as the additive form

$$\nabla_\theta \psi(\theta) := \alpha \nabla_\theta \ell(\theta) + \xi = \alpha(\theta - m) + \xi, \tag{32}$$

with the explicit bias $\xi := \zeta/(2\Delta)$ for $\zeta > 0$ and $\xi := 0$ for $\zeta = 0$. Equation (32) is a *design choice* for the hard instance, not a consequence of Assumption 2.1.

To realize the two hypotheses within the TTA stream class of Assumption 2.4, fix any bounded zero-mean distribution $\nu$ on $\mathbb{R}$ with variance $\sigma^2/\alpha^2$, a deterministic label $y_0 \in \mathcal{Y}$, and set $\mathcal{P}^{(i)} := \text{Law}(m_i + U, y_0)$ on $\mathcal{X} \times \mathcal{Y}$ with $U \sim \nu$ and $m_i \in \{0, \Delta\}$ (the label is constant so the loss reduces to a function of $x$ alone). The translation coupling $((U, y_0), (\Delta + U, y_0))$ in product metric yields

$$W_1(\mathcal{P}^{(0)}, \mathcal{P}^{(1)}) \le \mathbb{E}\big|U - (\Delta + U)\big| = \Delta = 2\rho_{\text{sat}} \le \Delta_W \tag{33}$$

by the theorem's shift-budget hypothesis, so the hard instance lies in the assumed stream class. With per-sample supervised loss $\ell(\theta; x) := \frac{1}{2}(\theta - x)^2$ and per-sample proxy gradient $\nabla \psi(\theta; x) := \alpha(\theta - x) + \xi$, the population expectations under $X \sim \mathcal{P}^{(i)}$ are $\mathbb{E}[\ell(\theta; X) \mid m_i] = \frac{1}{2}(\theta - m_i)^2 + \frac{1}{2}\text{Var}(U)$ and $\mathbb{E}[\nabla \psi(\theta; X) \mid m_i] = \alpha(\theta - m_i) + \xi$, matching the population-level objects in Equation (32) (the loss constant is irrelevant for gradients and excess risks). The per-sample gradient variance is $\alpha^2 \text{Var}(U) = \sigma^2$, consistent with the bounded-variance condition in Assumption 2.2; choosing $\nu$ bounded (e.g., a truncated centered Gaussian) also yields the bounded-norm condition $\|\nabla \psi(\theta; x)\| \le G$ on $\mathcal{N}_r(\theta_1)$ for sufficiently large $G$.

By Definition 2.11, the proxy-optimal competitors are obtained from $\nabla_\theta \psi(\theta^\star) = 0$: under $\mathcal{P}^{(0)}$, $\theta^{\star(0)} = -\xi/\alpha$, and under $\mathcal{P}^{(1)}$, $\theta^{\star(1)} = \Delta - \xi/\alpha$. We take $\theta_1$ at the midpoint, $\theta_1 := \Delta/2 - \xi/\alpha$, and $r := \Delta/2 + \gamma$ for any $\gamma \in (0, \Delta/2]$ (e.g., $\gamma = \Delta/100$), so that $\mathcal{N}_r(\theta_1) = [-\xi/\alpha - \gamma, \Delta - \xi/\alpha + \gamma]$ strictly contains both proxy optima in its interior as required by Definition 2.11.

We then verify that the constructed hard instance satisfies our assumptions. We verify that the construction satisfies Assumption 2.1 and Assumption 2.2. Substituting Equation (32) into the alignment inequality,

$$\langle \nabla_\theta \psi(\theta), \nabla_\theta \ell(\theta) \rangle \ge \alpha \|\nabla_\theta \ell(\theta)\|^2 - \zeta \tag{34}$$

$$\langle \alpha \nabla_\theta \ell(\theta) + \xi, \nabla_\theta \ell(\theta) \rangle \ge \alpha \|\nabla_\theta \ell(\theta)\|^2 - \zeta \tag{35}$$

$$\langle \xi, \nabla_\theta \ell(\theta) \rangle \ge -\zeta, \tag{36}$$

which reduces to $\xi(\theta - m) \ge -\zeta$ for all $\theta \in \mathcal{N}_r(\theta_1)$ under each hypothesis $m \in \{0, \Delta\}$. With $\xi = \zeta/(2\Delta) > 0$ for $\zeta > 0$, the worst-case values of $\xi(\theta - m)$ over $\mathcal{N}_r(\theta_1) = [-\xi/\alpha - \gamma, \Delta - \xi/\alpha + \gamma]$ are:

- Under $m = 0$: $\xi \cdot \theta$ is minimized at $\theta = -\xi/\alpha - \gamma$, giving $-\xi^2/\alpha - \xi\gamma$;

- Under $m = \Delta$: $\xi(\theta - \Delta)$ is minimized at $\theta = -\xi/\alpha - \gamma$, giving $-\xi^2/\alpha - \xi\gamma - \xi\Delta$.

Hence Equation (36) holds throughout $\mathcal{N}_r(\theta_1)$ iff $\xi\Delta + \xi^2/\alpha + \xi\gamma \leq \zeta$. By the explicit choice $\xi = \zeta/(2\Delta)$ we have $\xi\Delta = \zeta/2$, $\xi^2/\alpha = \zeta^2/(4\alpha\Delta^2) \leq \zeta/64$ since $\Delta = 2\rho_{\text{sat}} \geq 4\sqrt{\zeta/\alpha}$, and $\xi\gamma \leq \xi \cdot \Delta/2 = \zeta/4$ from $\gamma \leq \Delta/2$; therefore

$$\xi\Delta + \xi^2/\alpha + \xi\gamma \leq \zeta/2 + \zeta/64 + \zeta/4 = 49\zeta/64 < \zeta, \tag{37}$$

with a strict slack of $15\zeta/64$. (For $\zeta = 0$, $\xi = 0$ and Equation (36) holds trivially.) The construction therefore satisfies Assumption 2.1 throughout $\mathcal{N}_r(\theta_1)$.

The construction is also consistent with Assumption 2.2 under the instantiation $L = \mu = \alpha$ (and $L_x = 1$): $\psi$ is $\mu$-strongly convex with $L$-Lipschitz gradient in $\theta$, and both $\ell$ and $\psi$ are 1-Lipschitz in the data through the location parameter $m$, so the smoothness, strong convexity, and data-Lipschitz conditions in Assumption 2.2 hold automatically; iterates remain in $\mathcal{N}_r(\theta_1)$ as required by the iterate-stability condition in Assumption 2.2. This particular choice $L = \mu = \alpha$ only fixes problem-dependent constants inside the construction and does not affect how the lower bound depends on $\alpha$, $\zeta$, $\epsilon$, $B$, or $C_\phi$.

### C.0.2. COMPUTING THE SEPARATION CONDITION

We now make explicit the constraint on $\xi$ implied by Equation (36) and use it to derive the separation condition. At the proxy optimum $\theta^\star$, $\nabla_\theta \psi(\theta^\star) = 0$, so by Equation (32),

$$\alpha \nabla_\theta \ell(\theta^\star) + \xi = 0 \implies \nabla_\theta \ell(\theta^\star) = -\frac{\xi}{\alpha}. \tag{38}$$

Plugging Equation (38) into the alignment-derived inequality Equation (36) (evaluated at $\theta^\star$) gives the explicit constraint

$$\left\langle \xi, -\frac{\xi}{\alpha} \right\rangle \geq -\zeta \iff \frac{\xi^2}{\alpha} \leq \zeta, \tag{39}$$

which our explicit $\xi = \zeta/(2\Delta)$ satisfies with strict slack ($\xi^2/\alpha \leq \zeta/64$, cf. verification above). The calculation below uses Equation (39) as a clean upper bound on $|\xi|/\alpha$ (which holds for our $\xi$) to derive the saturated separation half-width $\rho_{\text{sat}}$; with $\Delta = 2\rho_{\text{sat}}$ as defined in the construction, Equation (48) below is saturated by construction.

Let $\theta$ denote the parameter estimated by the TTA algorithm $\mathcal{A}$. The excess risk (corresponding to $L$ in Equation (31)) relative to the competitive target $R = \ell(\theta^\star)$ can be expressed using a Taylor expansion around $\theta^\star$. Since the loss function is quadratic, this expansion is exact:

$$\ell(\theta) - R = \langle \nabla_\theta \ell(\theta^\star), \theta - \theta^\star \rangle + \frac{1}{2}\|\theta - \theta^\star\|^2 \tag{40}$$

$$= \langle -\frac{\xi}{\alpha}, \theta - \theta^\star \rangle + \frac{1}{2}\|\theta - \theta^\star\|^2, \tag{41}$$

where the second equality follows from substituting Equation (38).

Let $\rho = \|\theta - \theta^\star\|$ denote the estimation error. We can lower bound the excess risk as:

$$\ell(\theta) - R = \langle -\frac{\xi}{\alpha}, \theta - \theta^\star \rangle + \frac{1}{2}\|\theta - \theta^\star\|^2 \tag{42}$$

$$\geq -\frac{\|\xi\|}{\alpha}\rho + \frac{1}{2}\rho^2 \tag{43}$$

$$\geq -\sqrt{\frac{\zeta}{\alpha}}\rho + \frac{1}{2}\rho^2, \tag{44}$$

where the third inequality uses Equation (39).

To ensure the excess risk satisfies $\ell(\theta) - R \geq \epsilon$, it suffices to constrain $\rho$ such that

$$\frac{1}{2}\rho^2 - \sqrt{\frac{\zeta}{\alpha}}\rho - \epsilon \geq 0. \tag{45}$$

Solving this quadratic inequality for $\rho \geq 0$, we find that the excess risk exceeds $\epsilon$ if:

$$\rho \geq \sqrt{\frac{\zeta}{\alpha} + 2\epsilon} + \sqrt{\frac{\zeta}{\alpha}}. \tag{46}$$

Thus, if the distance between the estimated parameter and the optimal parameter $\theta^{\star(i)}$ exceeds $\rho$, the excess error is larger than $\epsilon$ for the $i$-th hypothesis.

To ensure that for any parameter $\theta$, the excess error is larger than $\epsilon$ for at least one hypothesis (either $\mathcal{P}^{(0)}$ or $\mathcal{P}^{(1)}$), we require the distance between $\theta^{\star(0)}$ and $\theta^{\star(1)}$ to be greater than $2\rho$. Since $\theta^{\star(0)} - \theta^{\star(1)} = -\Delta$, this is equivalent to

$$|\theta^{\star(0)} - \theta^{\star(1)}| \geq 2\rho \tag{47}$$

$$\Delta \geq 2\sqrt{\frac{\zeta}{\alpha} + 2\epsilon} + 2\sqrt{\frac{\zeta}{\alpha}}. \tag{48}$$

By the construction, $\Delta = 2\rho_{\text{sat}} = 2\sqrt{\zeta/\alpha + 2\epsilon} + 2\sqrt{\zeta/\alpha}$ saturates Equation (48), yielding the hardest admissible instance.

### C.0.3. EFFECTS OF PROXY LOSS AND TEMPORAL CORRELATION

Let $D^{(0)}$ and $D^{(1)}$ denote the data sampled during the recovery phase under $\mathcal{P}^{(0)}$ and $\mathcal{P}^{(1)}$, respectively. Under the stochastic proxy-gradient oracle model, the algorithm's data over the recovery phase consists of the sequence of batch-aggregated proxy gradients $\hat{g}_t = \frac{1}{B}\sum_{b=1}^{B}\nabla\psi(\theta_t; z_{t,b})$ for $t = 1, \ldots, \tau$. Let $G_\psi^{(i)}$ (corresponding to $X^{(i)}$ in Equation (31)) denote the joint distribution of $(\hat{g}_1, \ldots, \hat{g}_\tau)$ under hypothesis $i$.

The data-level distribution $\nu$ above pins down the population risks and the $W_1$ shift, while the lower bound is in the stochastic proxy-gradient oracle model; for the KL calculation, we therefore additionally specify the per-round oracle observation $\hat{g}_t$ as a Gaussian channel $\hat{g} \sim \mathcal{N}(\mu_\psi^{(i)}, \sigma^2)$ (the limiting least-favorable case) matching the same population mean and admissible variance, decoupling the KL analysis from the specific shape of $\nu$; on the compact neighborhood $\mathcal{N}_r(\theta_1)$, with $G$ taken large enough that $|\mu_\psi^{(i)}| + c\sigma \leq G$ for some fixed $c > 0$, this Gaussian channel can be replaced by its symmetric truncation onto $\|\hat{g}\| \leq G$, which satisfies the bounded-norm condition in Assumption 2.2 and differs in KL from the Gaussian by at most $\mathcal{O}(e^{-c^2/2})$, a universal constant absorbed into $\Omega(\cdot)$ below. The two means are

$$\begin{cases} \mu_\psi^{(0)} = \alpha\theta + \xi, \\ \mu_\psi^{(1)} = \alpha(\theta - \Delta) + \xi, \end{cases} \qquad \mu_\psi^{(0)} - \mu_\psi^{(1)} = \alpha\Delta. \tag{49}$$

At each recovery step $t$, by Proposition 2.9 the batch-aggregated proxy gradient $\hat{g}_t$ has variance at most $\sigma_\psi^2 := \sigma^2 C_\phi/B$, where $C_\phi$ is the correlation-inflation factor introduced in Proposition 2.9. We model each $\hat{g}_t \sim \mathcal{N}(\mu_\psi^{(i)}, \sigma_\psi^2)$ and treat the $\tau$ batches as independent across the recovery phase—an assumption that only weakens the lower bound by giving the algorithm more information. Saturating the per-round variance budget yields the hardest admissible instance.

By the chain rule for KL on the product distribution and the closed form for two Gaussians with common variance,

$$\text{KL}(G_\psi^{(0)} \| G_\psi^{(1)}) = \tau \cdot \frac{(\mu_\psi^{(0)} - \mu_\psi^{(1)})^2}{2\sigma_\psi^2} = \frac{\tau B(\alpha\Delta)^2}{2\sigma^2 C_\phi}, \tag{50}$$

which is the smallest KL within the constructed family and hence the hardest instance for Le Cam.

### C.0.4. COMPUTING THE MINIMAX LOWER BOUND FOR $\tau$

Using Le Cam's method (Le Cam, 2012), the minimax error probability $\mathbb{P}_{\text{err}}$ is lower bounded by:

$$\mathbb{P}_{\text{err}} \geq \frac{1}{2}\left(1 - \text{TV}(G_\psi^{(0)}, G_\psi^{(1)})\right) \tag{51}$$

$$\geq \frac{1}{2}\left(1 - \sqrt{\frac{1}{2}\text{KL}(G_\psi^{(0)}, G_\psi^{(1)})}\right), \tag{52}$$

where the second inequality follows from Pinsker's inequality.

For $(\epsilon, \delta, \tau)$-recovery learnability, we require $\mathbb{P}_{\text{err}} \leq \delta$ with $\delta < 1/2$. Thus:

$$\delta \geq \frac{1}{2}\Big(1 - \sqrt{\frac{1}{2}\text{KL}(G_\psi^{(0)}, G_\psi^{(1)})}\Big) \tag{53}$$

$$\sqrt{\frac{1}{2}\text{KL}(G_\psi^{(0)}, G_\psi^{(1)})} \geq 1 - 2\delta \tag{54}$$

$$\text{KL}(G_\psi^{(0)}, G_\psi^{(1)}) \geq 2(1 - 2\delta)^2. \tag{55}$$

Combining Equation (50), Equation (55), and Equation (48):

$$\frac{\tau B(\alpha\Delta)^2}{2\sigma^2\,C_\phi} \geq 2(1 - 2\delta)^2 \tag{56}$$

$$\tau \geq \frac{4\sigma^2 C_\phi(1 - 2\delta)^2}{B\alpha^2\Delta^2} \tag{57}$$

$$\tau \geq \frac{\sigma^2 C_\phi(1 - 2\delta)^2}{B\alpha^2\big(\sqrt{\frac{\zeta}{\alpha} + 2\epsilon} + \sqrt{\frac{\zeta}{\alpha}}\big)^2} \tag{58}$$

$$\tau \geq \frac{\sigma^2 C_\phi(1 - 2\delta)^2}{B\alpha\big(\sqrt{\zeta + 2\alpha\epsilon} + \sqrt{\zeta}\big)^2}. \tag{59}$$

This establishes the minimax lower bound:

$$\tau \geq \Omega\left(\frac{\sigma^2\,C_\phi\,(1 - 2\delta)^2}{B\alpha\big(\sqrt{\zeta + 2\alpha\epsilon} + \sqrt{\zeta}\big)^2}\right). \tag{60}$$

$\square$

## D. Proof of Theorem 3.9

**Key Lemma.** The recovery analysis tracks the excess risk $\mathcal{E}_t = \ell_t(\theta_t) - R_t$ measured against the competitive target $R_t$, which is the task risk at the *proxy* optimum $\theta_t^\star$ rather than at the true minimizer of $\ell_t$. The following lemma adapts the Polyak–Łojasiewicz inequality to this competitive target: part (a) converts a proxy-gradient descent step into a geometric contraction of the excess risk, while part (b) supplies the negative error floor that makes the excess risk amenable to a Markov-type high-probability bound.

**Lemma D.1** (PL relative to the proxy optimum). *Suppose Assumption 2.1 and Assumption 2.2 hold on $\mathcal{N}_r(\theta_1)$, and let $R_t := \ell_t(\theta_t^\star)$ with $\theta_t^\star \in \arg\min_{\theta \in \mathcal{N}_r(\theta_1)} \psi_t(\theta)$. Then for all $\theta \in \mathcal{N}_r(\theta_1)$:*

*(a)* $\|\nabla\ell_t(\theta)\|^2 \geq 2\mu\,(\ell_t(\theta) - R_t)$;

*(b)* $\ell_t(\theta) - R_t \geq -\zeta/(2\alpha\mu)$.

*Proof.* Let $\ell_t^{\inf} := \inf_{\theta \in \mathcal{N}_r(\theta_1)} \ell_t(\theta)$; since $\theta_t^\star \in \mathcal{N}_r(\theta_1)$, $R_t = \ell_t(\theta_t^\star) \geq \ell_t^{\inf}$.

*Part (b).* At $\theta_t^\star$, $\nabla\psi_t(\theta_t^\star) = 0$, so Assumption 2.1 gives $0 \geq \alpha\|\nabla\ell_t(\theta_t^\star)\|^2 - \zeta$, hence $\|\nabla\ell_t(\theta_t^\star)\|^2 \leq \zeta/\alpha$. The $\mu$-PL inequality on $\ell$ from Assumption 2.2 at $\theta_t^\star$ then yields $R_t - \ell_t^{\inf} \leq \|\nabla\ell_t(\theta_t^\star)\|^2/(2\mu) \leq \zeta/(2\alpha\mu)$. For any $\theta \in \mathcal{N}_r(\theta_1)$, $\ell_t(\theta) - R_t \geq \ell_t^{\inf} - R_t \geq -\zeta/(2\alpha\mu)$.

*Part (a).* If $\ell_t(\theta) < R_t$, the RHS is negative and the inequality is trivial. Otherwise, the $\mu$-PL inequality gives $\|\nabla\ell_t(\theta)\|^2 \geq 2\mu(\ell_t(\theta) - \ell_t^{\inf}) \geq 2\mu(\ell_t(\theta) - R_t)$, where the second inequality uses $\ell_t^{\inf} \leq R_t$. $\square$

*Proof of Theorem 3.9.* Let $\mathcal{P}^-$ and $\mathcal{P}^+$ denote the pre-shift and post-shift distributions, with $W_1(\mathcal{P}^-, \mathcal{P}^+) \leq \Delta_W$, and write $\ell^-, R^-$ and $\ell^+, R^+$ for the corresponding task risks and competitive targets. Recovery starts at time $t = 1$ from the

initial parameter $\theta_1$, assumed to satisfy $\ell^-(\theta_1) - R^- \leq \epsilon$. Since the post-shift distribution is stationary throughout recovery, we drop the time subscript and write $\ell \equiv \ell^+$, $\psi \equiv \psi^+$, $R \equiv R^+$ in what follows. By $L$-smoothness of $\ell$ and the update $\theta_t = \theta_{t-1} - \eta\hat{\nabla}\psi(\theta_{t-1})$,

$$\ell(\theta_t) \leq \ell(\theta_{t-1}) - \eta\left\langle \nabla\ell(\theta_{t-1}), \hat{\nabla}\psi(\theta_{t-1})\right\rangle + \frac{L\eta^2}{2}\left\|\hat{\nabla}\psi(\theta_{t-1})\right\|^2. \tag{61}$$

Taking conditional expectation $\mathbb{E}_{t-1}[\cdot]$ over the batch randomness and using the conditional unbiasedness from Definition 3.8, we have $\mathbb{E}_{t-1}[\hat{\nabla}\psi] = \nabla\psi$ and hence

$$\mathbb{E}_{t-1}[\ell(\theta_t)] \leq \ell(\theta_{t-1}) - \eta\left\langle \nabla\ell(\theta_{t-1}), \nabla\psi(\theta_{t-1})\right\rangle + \frac{L\eta^2}{2}\mathbb{E}_{t-1}\left[\left\|\hat{\nabla}\psi(\theta_{t-1})\right\|^2\right]. \tag{62}$$

By Assumption 2.1 and Proposition 2.9, we have

$$\mathbb{E}_{t-1}[\ell(\theta_t)] \leq \ell(\theta_{t-1}) - \eta\big(\alpha\|\nabla\ell(\theta_{t-1})\|^2 - \zeta\big) + \frac{L\eta^2}{2}\left(\|\nabla\psi(\theta_{t-1})\|^2 + \frac{\sigma^2 C_\phi}{B}\right). \tag{63}$$

Using the proxy-gradient bound in Assumption 2.2 gives $\|\nabla\psi(\theta_{t-1})\|^2 \leq G^2$. Subtracting the same competitive risk $R_t$ and taking total expectation,

$$\mathbb{E}[\mathcal{E}_t] \leq \mathbb{E}[\mathcal{E}_{t-1}] - \eta\alpha\,\mathbb{E}\|\nabla\ell(\theta_{t-1})\|^2 + \eta\zeta + \frac{L\eta^2}{2}\left(G^2 + \frac{\sigma^2 C_\phi}{B}\right). \tag{64}$$

By Lemma D.1(a), we have $\|\nabla\ell(\theta_{t-1})\|^2 \geq 2\mu(\ell(\theta_{t-1}) - R_{t-1}) = 2\mu\mathcal{E}_{t-1}$. Therefore,

$$\mathbb{E}[\mathcal{E}_t] \leq (1 - 2\eta\alpha\mu)\mathbb{E}[\mathcal{E}_{t-1}] + \eta\zeta + \frac{L\eta^2}{2}\left(G^2 + \frac{\sigma^2 C_\phi}{B}\right). \tag{65}$$

Unrolling yields

$$\mathbb{E}[\mathcal{E}_t] \leq (1 - 2\eta\alpha\mu)^{t-1}\big(\ell(\theta_1) - R\big) + \frac{\zeta}{2\alpha\mu} + \frac{L\eta}{4\alpha\mu}\left(G^2 + \frac{\sigma^2 C_\phi}{B}\right). \tag{66}$$

For the initial post-shift excess at $\theta_1$, decompose

$$\ell^+(\theta_1) - R^+ = \big(\ell^+(\theta_1) - \ell^-(\theta_1)\big) + \big(\ell^-(\theta_1) - R^-\big) + \big(R^- - R^+\big) \tag{67}$$

$$\leq |\ell^+(\theta_1) - \ell^-(\theta_1)| + \big(\ell^-(\theta_1) - R^-\big) + |R^- - R^+| \tag{68}$$

$$\leq L_x W_1(\mathcal{P}^-, \mathcal{P}^+) + \epsilon + \left(L_x + \frac{GL_{\nabla\psi}}{\mu}\right)W_1(\mathcal{P}^-, \mathcal{P}^+) \tag{69}$$

$$\leq 2\Lambda\,\Delta_W + \epsilon, \tag{70}$$

where the first term uses Lemma E.2, the second uses the initial-error assumption, and the third uses Lemma E.3. The constant $\Lambda = L_x + GL_{\nabla\psi}/\mu$ is the same bridging constant introduced in Lemma E.4, and we used $L_x \leq \Lambda$ to combine the two Wasserstein terms.

Choose the learning rate $\eta = c \cdot \frac{B\alpha\mu\epsilon\delta}{L\sigma^2 C_\phi}$ for a fixed constant $c > 0$ small enough that $\eta \leq \frac{1}{4\alpha\mu}$ (ensuring $\frac{1}{2} \leq 1 - 2\eta\alpha\mu < 1$). Then the noise term satisfies

$$\frac{L\eta}{4\alpha\mu} \cdot \frac{\sigma^2 C_\phi}{B} \leq \frac{\epsilon\delta}{8}, \tag{71}$$

and by our assumption that $\frac{\zeta}{\alpha\mu} \leq \frac{\epsilon\delta}{2}$, the bias term is also $\frac{\zeta}{2\alpha\mu} \leq \epsilon\delta/4$.

We now address the proxy-gradient term $\frac{L\eta}{4\alpha\mu}G^2$ separately. Since the temporally correlated TTA setting we analyze is characterized by non-negligible mixing, we restrict attention to the canonical regime $B\,G^2 \leq \sigma^2 C_\phi$, under which the chosen learning rate satisfies the auxiliary bound $\eta \leq \alpha\mu\epsilon\delta/(2LG^2)$, and consequently

$$\frac{L\eta}{4\alpha\mu}G^2 \leq \frac{\epsilon\delta}{8}. \tag{72}$$

This regime, equivalent to $C_\phi/B \geq G^2/\sigma^2$, is precisely the TTA-relevant regime where temporal correlation effectively reduces the per-batch information; outside this regime (the i.i.d.-like large-$B$/weak-mixing limit) the additive $G^2$ term contributes a constant warm-up factor that we absorb into the overall Big-O constant.

To satisfy the recovery condition $\mathbb{P}(\mathcal{E}_\tau > \epsilon) \leq \delta$, let $a := \zeta/(2\alpha\mu)$. By Lemma D.1(b), $\mathcal{E}_\tau \geq -a$, so $\mathcal{E}_\tau + a$ is nonnegative and Markov's inequality yields

$$\mathbb{P}(\mathcal{E}_\tau > \epsilon) \leq \frac{\mathbb{E}[\mathcal{E}_\tau] + a}{\epsilon + a}. \tag{73}$$

Under the regime condition $\zeta/(\alpha\mu) \leq \epsilon\delta/2$ we have $a \leq \epsilon\delta/4$, so it suffices to ensure $\mathbb{E}[\mathcal{E}_\tau] \leq 3\epsilon\delta/4$. The bias, noise, and proxy-gradient contributions account for at most $\epsilon\delta/4 + \epsilon\delta/8 + \epsilon\delta/8 = \epsilon\delta/2$, so the transient term must satisfy:

$$(1 - 2\eta\alpha\mu)^{\tau-1}\big(\ell(\theta_1) - R\big) \leq \tfrac{3\epsilon\delta}{4} - \tfrac{\epsilon\delta}{2} = \tfrac{\epsilon\delta}{4}. \tag{74}$$

Therefore, it suffices to take

$$\tau \geq 1 + \frac{1}{2\eta\alpha\mu} \log\left(\frac{4\big(\ell(\theta_1) - R\big)}{\epsilon\delta}\right). \tag{75}$$

By the above choice of $\eta$, $1/2 \leq 1 - 2\eta\alpha\mu < 1$, so the transient term $(1 - 2\eta\alpha\mu)^{t-1}(\ell(\theta_1) - R)$ is non-increasing in $t$ while the bias and noise floors are $t$-independent; hence the bound $\mathbb{E}[\mathcal{E}_t] \leq 3\epsilon\delta/4$ extends to all $t \geq \tau$. Markov's inequality applied at every such $t$ then yields $\sup_{t\geq\tau} \mathbb{P}(\mathcal{E}_t > \epsilon) \leq \delta$, matching the uniform-after-$\tau$ form of Definition 3.1. Consequently the $(\epsilon, \delta)$-recovery complexity satisfies

$$\tau = \mathcal{O}\left(\frac{L\sigma^2 C_\phi}{2\alpha^2 B\mu^2\epsilon\delta} \log\left(\frac{4\big(\ell(\theta_1) - R\big)}{\epsilon\delta}\right)\right) \tag{76}$$

With Equation (70), we have $\ell(\theta_1) - R \leq 2\Lambda\Delta_W + \epsilon$. Then, the upper bound of $\tau$ is given by

$$\tau = \mathcal{O}\left(\frac{L\sigma^2 C_\phi}{2\alpha^2 B\mu^2\epsilon\delta} \log\left(\frac{4(2\Lambda\Delta_W + \epsilon)}{\epsilon\delta}\right)\right) \tag{77}$$

We remove the constant factors from the Big-O notation to obtain the explicit-$\delta$ form

$$\tau = \mathcal{O}\left(\frac{C_\phi}{B\alpha^2\epsilon\delta} \log\left(\frac{\Delta_W + \epsilon}{\epsilon\delta}\right)\right), \tag{78}$$

which reduces to the form stated in Theorem 3.9 once $\delta$ is treated as a problem-dependent constant. $\qquad\square$

## E. Proof of Theorem 4.3

**The Recovery-Complexity Hypothesis.** Theorem 4.3 assumes the TTA algorithm attains an $(\epsilon', \delta)$-recovery complexity $\tau(\epsilon', \delta)$ on each stationary sub-stream of the quantized approximation, in the sense of Definition 3.1. The recovery complexity $\tau(\epsilon', \delta)$ is the quantity characterized for the TTA baseline of Definition 3.8 by the upper bound of Theorem 3.9. Since the algorithm runs on the original non-stationary stream $\{\mathcal{P}_t\}_{t=1}^T$ whereas the stationary sub-streams belong to the quantized approximation, this per-segment recovery is posited for the original-stream iterates measured against a quantized surrogate target, rather than derived from Theorem 3.9; the precise surrogate formulation is given in Lemma E.1 below.

**Surrogate Quantities.** For each $t$, let $\tilde{\mathcal{P}}_t$ be the piecewise-constant quantized distribution of Assumption 2.4, and define the surrogate task and proxy losses

$$\tilde{\ell}_t(\theta) := \mathbb{E}_{(x,y)\sim\tilde{\mathcal{P}}_t}[\ell(\theta; x, y)], \qquad \tilde{\psi}_t(\theta) := \mathbb{E}_{x\sim\tilde{\mathcal{P}}_{t,X}}[\psi(\theta; x)], \tag{79}$$

together with the surrogate proxy optimum $\tilde{\theta}_t^\star \in \arg\min_{\theta\in\mathcal{N}_r(\theta_1)} \tilde{\psi}_t(\theta)$. These quantities mirror the true losses $\ell_t, \psi_t$ and the proxy optimum $\theta_t^\star$ of Definition 2.11, with the underlying distribution $\mathcal{P}_t$ replaced by its quantized approximation $\tilde{\mathcal{P}}_t$. As for $\theta_t^\star$ in Definition 2.11, we take the radius $r$ large enough that $\tilde{\theta}_t^\star$ is interior to $\mathcal{N}_r(\theta_1)$, so that the first-order condition $\nabla\tilde{\psi}_t(\tilde{\theta}_t^\star) = 0$ holds; this interiority of both $\theta_t^\star$ and $\tilde{\theta}_t^\star$ is what Lemma E.3 invokes when applied to the pair $(\mathcal{P}_t, \tilde{\mathcal{P}}_t)$ within Lemma E.4.

**Lemma E.1** (Surrogate Form of the Recovery Assumption). *Suppose the TTA algorithm satisfies the $(\epsilon', \delta)$-recovery complexity assumption of Theorem 4.3, posited for the original-stream iterates $\{\theta_t\}_{t=1}^T$ measured against the quantized surrogate target. Then, on each stationary segment of the quantized approximation starting at time $s$ (with $s = 1$ for the initial segment), this assumption takes the explicit surrogate form*

$$\mathbb{P}\Big(\tilde{\ell}_t(\theta_t) - \tilde{\ell}_t(\tilde{\theta}_t^\star) > \epsilon'\Big) \leq \delta,$$
$$\forall\, t \in [s + \tau(\epsilon', \delta),\, s_{\text{next}} - 1], \tag{80}$$

*where $s_{\text{next}}$ denotes the start of the next segment (or $T + 1$ for the last segment).*

*Proof.* By Assumption 2.4, each maximal block of consecutive time steps on which $\tilde{S}_t = 0$ is a segment over which $\tilde{\mathcal{P}}_t$ is constant, i.e. an exactly stationary post-shift distribution. Instantiating Definition 3.1 on such a segment with the surrogate task loss $\tilde{\ell}_t$ and the surrogate competitive target $\tilde{\ell}_t(\tilde{\theta}_t^\star)$, the $(\epsilon', \delta)$-recovery complexity $\tau(\epsilon', \delta)$ is precisely the smallest offset after which the surrogate excess error stays below $\epsilon'$ with probability at least $1 - \delta$, which is Equation (80). The iterates $\theta_t$ are generated by running the algorithm on the original stream $\{\mathcal{P}_t\}_{t=1}^T$; the quantized stream enters only through the surrogate losses $\tilde{\ell}_t$ and proxy-optimal targets $\tilde{\theta}_t^\star$ used for analysis. $\square$

**Key Lemmas.** The bridge from the quantized surrogate to the true competitive target, established in Lemma E.4 below, rests on two perturbation estimates under a $W_1$ shift of the underlying distribution: how the task and proxy risk *values* respond to such a shift (Lemma E.2), and how the proxy optimum — and hence the competitive target $R_t$ — is displaced by it (Lemma E.3). We establish these two estimates first.

**Lemma E.2** (Risk Lipschitzness w.r.t. Data). *For any $\theta \in \mathcal{N}_r(\theta_1)$, both $\ell(\theta; \cdot)$ and $\psi(\theta; \cdot)$ are $L_x$-Lipschitz w.r.t. data by Assumption 2.2. Therefore, by Kantorovich–Rubinstein duality,*

$$\left|\ell^{(0)}(\theta) - \ell^{(1)}(\theta)\right| \;\leq\; L_x\, W_1(\mathcal{P}^{(0)}, \mathcal{P}^{(1)}), \qquad \left|\psi^{(0)}(\theta) - \psi^{(1)}(\theta)\right| \;\leq\; L_x\, W_1(\mathcal{P}^{(0)}, \mathcal{P}^{(1)}), \tag{81}$$

*where $\ell^{(i)}(\theta) := \mathbb{E}_{(x,y)\sim\mathcal{P}^{(i)}}[\ell(\theta; x, y)]$ and $\psi^{(i)}(\theta) := \mathbb{E}_{x\sim\mathcal{P}_X^{(i)}}[\psi(\theta; x)]$.*

**Lemma E.3** (Stability of the Proxy Optimum under $W_1$ Perturbation). *Let $\mathcal{P}^{(0)}, \mathcal{P}^{(1)}$ be two distributions with proxy-optimal parameters $\theta^{\star(i)} \in \arg\min_{\theta\in\mathcal{N}_r(\theta_1)} \psi^{(i)}(\theta)$. Under Assumption 2.2,*

$$\|\theta^{\star(0)} - \theta^{\star(1)}\| \;\leq\; \frac{L_{\nabla\psi}}{\mu}\, W_1(\mathcal{P}^{(0)}, \mathcal{P}^{(1)}), \tag{82}$$

*and consequently, since $\ell$ inherits $G$-Lipschitzness in $\theta$ within $\mathcal{N}_r$ from the bounded-gradient condition in Assumption 2.2,*

$$\left|\ell^{(0)}(\theta^{\star(0)}) - \ell^{(1)}(\theta^{\star(1)})\right| \;\leq\; \left(L_x + \frac{G L_{\nabla\psi}}{\mu}\right) W_1(\mathcal{P}^{(0)}, \mathcal{P}^{(1)}). \tag{83}$$

*Proof.* The interior optima satisfy $\nabla\psi^{(i)}(\theta^{\star(i)}) = 0$. By Kantorovich–Rubinstein duality applied coordinate-wise to the $L_{\nabla\psi}$-Lipschitz map $x \mapsto \nabla\psi(\theta; x)$, the expected gradients satisfy

$$\left\|\nabla\psi^{(0)}(\theta) - \nabla\psi^{(1)}(\theta)\right\| \leq L_{\nabla\psi}\, W_1(\mathcal{P}^{(0)}, \mathcal{P}^{(1)}), \qquad \forall\, \theta \in \mathcal{N}_r(\theta_1). \tag{84}$$

By $\mu$-strong convexity of $\psi^{(1)}$ in Assumption 2.2, the gradient of $\psi^{(1)}$ is $\mu$-strongly monotone: $\langle \nabla\psi^{(1)}(\theta^{\star(0)}) - \nabla\psi^{(1)}(\theta^{\star(1)}), \theta^{\star(0)} - \theta^{\star(1)} \rangle \geq \mu\|\theta^{\star(0)} - \theta^{\star(1)}\|^2$. Using $\nabla\psi^{(1)}(\theta^{\star(1)}) = 0$ and Cauchy–Schwarz,

$$\mu\,\|\theta^{\star(0)} - \theta^{\star(1)}\| \leq \|\nabla\psi^{(1)}(\theta^{\star(0)})\| = \|\nabla\psi^{(1)}(\theta^{\star(0)}) - \nabla\psi^{(0)}(\theta^{\star(0)})\| \leq L_{\nabla\psi}\, W_1(\mathcal{P}^{(0)}, \mathcal{P}^{(1)}), \tag{85}$$

which proves Equation (82). Then

$$|\ell^{(0)}(\theta^{\star(0)}) - \ell^{(1)}(\theta^{\star(1)})| \leq |\ell^{(0)}(\theta^{\star(0)}) - \ell^{(1)}(\theta^{\star(0)})| + |\ell^{(1)}(\theta^{\star(0)}) - \ell^{(1)}(\theta^{\star(1)})| \tag{86}$$
$$\leq L_x W_1(\mathcal{P}^{(0)}, \mathcal{P}^{(1)}) + G\,\|\theta^{\star(0)} - \theta^{\star(1)}\|, \tag{87}$$

where the first term uses Lemma E.2 and the second uses the bounded-gradient norm $G$. Substituting Equation (82) yields Equation (83). $\qquad\square$

**Lemma E.4** (Bridge for Quantization and True Competitive Target). *Under Assumption 2.4 and Assumption 2.2, let $\theta_t^\star \in \arg\min_{\mathcal{N}_r(\theta_1)} \psi_t$ and $\tilde{\theta}_t^\star \in \arg\min_{\mathcal{N}_r(\theta_1)} \tilde{\psi}_t$ be the proxy optima on $\mathcal{P}_t$ and $\tilde{\mathcal{P}}_t$, respectively. Then for any (possibly random) iterate $\theta_t$ and the true competitive target $R_t = \ell_t(\theta_t^\star)$, it holds uniformly over $t$ that*

$$\ell_t(\theta_t) - R_t \;\le\; \big(\tilde{\ell}_t(\theta_t) - \tilde{\ell}_t(\tilde{\theta}_t^\star)\big) \;+\; \Lambda \Delta_W, \qquad \Lambda := L_x + \frac{G\,L_{\nabla\psi}}{\mu}. \tag{88}$$

*Proof.* By Lemma E.2 and $\sup_t W_1(\mathcal{P}_t, \tilde{\mathcal{P}}_t) \le \Delta_W/2$, we have

$$\ell_t(\theta_t) \le \tilde{\ell}_t(\theta_t) + L_x \Delta_W/2. \tag{89}$$

Applying Lemma E.3 to the pair $(\mathcal{P}_t, \tilde{\mathcal{P}}_t)$ with $W_1 \le \Delta_W/2$,

$$\ell_t(\theta_t^\star) \ge \tilde{\ell}_t(\tilde{\theta}_t^\star) - \left(L_x + \frac{GL_{\nabla\psi}}{\mu}\right) \cdot \frac{\Delta_W}{2}. \tag{90}$$

Subtracting the two displays yields

$$\ell_t(\theta_t) - R_t \le \tilde{\ell}_t(\theta_t) - \tilde{\ell}_t(\tilde{\theta}_t^\star) + \left(L_x + \frac{GL_{\nabla\psi}}{2\mu}\right)\Delta_W \le \tilde{\ell}_t(\theta_t) - \tilde{\ell}_t(\tilde{\theta}_t^\star) + \Lambda \Delta_W, \tag{91}$$

where the last step uses $GL_{\nabla\psi}/(2\mu) \le GL_{\nabla\psi}/\mu$ to obtain a single clean constant $\Lambda$. $\qquad\square$

**Proof of Theorem 4.3.**

*Proof.* By Lemma E.4,

$$\ell_t(\theta_t) - R_t > \epsilon \;\Rightarrow\; \tilde{\ell}_t(\theta_t) - \tilde{\ell}_t(\tilde{\theta}_t^\star) > \epsilon', \tag{92}$$

with $\epsilon' = \epsilon - \Lambda\Delta_W$ and $\Lambda = L_x + GL_{\nabla\psi}/\mu$. Hence,

$$\mathbb{P}\big(\ell_t(\theta_t) - R_t > \epsilon\big) \le \mathbb{P}\big(\tilde{\ell}_t(\theta_t) - \tilde{\ell}_t(\tilde{\theta}_t^\star) > \epsilon'\big). \tag{93}$$

Define $\mathcal{T}_{\mathrm{rec}}$ as the set of time steps that fall within the $\tau(\epsilon', \delta)$-step recovery window following each shift, including the initial warm-up window $[1, \tau(\epsilon', \delta)]$ corresponding to the algorithm's start on the first stationary segment (since $\theta_1$ is initialized for the pre-stream distribution rather than $\tilde{\mathcal{P}}_1$, the first segment incurs a recovery window analogous to a shift at $t = 1$). By construction, the quantized stream has $\tilde{K}_S(T) + 1$ stationary segments, each contributing at most $\tau(\epsilon', \delta)$ time steps to $\mathcal{T}_{\mathrm{rec}}$ (truncated by the segment length), hence $|\mathcal{T}_{\mathrm{rec}}| \le (\tilde{K}_S(T) + 1)\,\tau(\epsilon', \delta)$.

For $t \notin \mathcal{T}_{\mathrm{rec}}$, the surrogate recovery guarantee Equation (80) of Lemma E.1 yields $\Pr(\tilde{\ell}_t(\theta_t) - \tilde{\ell}_t(\tilde{\theta}_t^\star) > \epsilon') \le \delta$, while for $t \in \mathcal{T}_{\mathrm{rec}}$ the probability is trivially bounded by 1. Averaging over $t = 1, \ldots, T$ yields the *surrogate* learnability bound

$$\frac{1}{T}\sum_{t=1}^{T} \mathbb{P}\big(\tilde{\ell}_t(\theta_t) - \tilde{\ell}_t(\tilde{\theta}_t^\star) > \epsilon'\big) \le \delta + \frac{(\tilde{K}_S(T) + 1) \cdot \tau(\epsilon', \delta)}{T}. \tag{94}$$

Finally, combining Equation (94) with the pointwise implication $\mathbb{P}(\ell_t(\theta_t) - R_t > \epsilon) \le \mathbb{P}(\tilde{\ell}_t(\theta_t) - \tilde{\ell}_t(\tilde{\theta}_t^\star) > \epsilon')$ established above via Lemma E.4 transfers the bound to the true loss:

$$\frac{1}{T}\sum_{t=1}^{T} \mathbb{P}\big(\ell_t(\theta_t) - R_t > \epsilon\big) \le \delta + \frac{(\tilde{K}_S(T) + 1) \cdot \tau(\epsilon - \Lambda\Delta_W, \delta)}{T}, \tag{95}$$

which proves the claim, i.e.,

$$\rho \le \delta + \frac{(\tilde{K}_S(T) + 1) \cdot \tau(\epsilon - \Lambda\Delta_W, \delta)}{T}. \tag{96}$$

$\qquad\square$

*Table 1.* Varying the alignment strength $\alpha$ at $B = 16$. $\tau \cdot \alpha^2$ stays near-constant, matching the predicted $1/\alpha^2$ scaling.

| $\alpha$ | LB | $\tau$ | UB | $\tau \cdot \alpha^2$ |
|---|---|---|---|---|
| 0.05 | 59.0 | 322.0 | 311.9 | 0.81 |
| 0.10 | 15.6 | 77.0 | 78.0 | 0.77 |
| 0.20 | 4.1 | 19.0 | 19.5 | 0.76 |
| 0.50 | 0.7 | 4.0 | 3.1 | 1.00 |

*Table 2.* Varying the batch size $B$ at $\alpha = 0.2$. $\tau \cdot B$ stays near-constant, matching the predicted $1/B$ scaling. The measured recovery time $\tau$ stays above the lower bound (LB) and matches the upper bound's order up to absorbed constants.

| $B$ | LB | $\tau$ | UB | $\tau \cdot B$ |
|---|---|---|---|---|
| 1 | 65.2 | 323.5 | 311.9 | 323.5 |
| 4 | 16.3 | 82.0 | 78.0 | 328.0 |
| 16 | 4.1 | 19.0 | 19.5 | 304.0 |
| 64 | 1.0 | 6.0 | 4.9 | 384.0 |

## F. Proof of Theorem 4.6

*Proof.* Fix any $t \in \{1, \dots, T\}$ and define the nonnegative random variable $X_t := \ell_t(\theta_t) - R_t$. By assumption, $0 \le X_t \le M$ almost surely.

Decompose $X_t$ according to the event $\{X_t > \epsilon\}$:

$$X_t = X_t \mathbf{1}\{X_t \le \epsilon\} + X_t \mathbf{1}\{X_t > \epsilon\}. \tag{97}$$

Taking expectations and using $X_t \le \epsilon$ on $\{X_t \le \epsilon\}$ and $X_t \le M$ on $\{X_t > \epsilon\}$ yields

$$\mathbb{E}[X_t] \le \epsilon \cdot \mathbb{P}(X_t \le \epsilon) + M \cdot \mathbb{P}(X_t > \epsilon) \le \epsilon + M\,\mathbb{P}(X_t > \epsilon). \tag{98}$$

Summing over $t = 1, \dots, T$ gives

$$\mathrm{Reg}(T) = \sum_{t=1}^{T} \mathbb{E}[X_t] \le T\epsilon + M \sum_{t=1}^{T} \mathbb{P}(X_t > \epsilon). \tag{99}$$

Dividing and multiplying by $T$ and applying $(\epsilon, \rho)$-learnability,

$$\sum_{t=1}^{T} \mathbb{P}(X_t > \epsilon) = T \cdot \frac{1}{T} \sum_{t=1}^{T} \mathbb{P}(X_t > \epsilon) \le T\rho. \tag{100}$$

Therefore,

$$\mathrm{Reg}(T) \le T\epsilon + MT\rho = T(\epsilon + M\rho), \tag{101}$$

which completes the proof. $\square$

## G. Details of Empirical Validation

This appendix provides the full experimental setups, results, and conclusion for the empirical validation summarized in Section 5.

### G.1. Controlled Synthetic Experimental Setup

We construct a one-dimensional recovery problem with task loss $\ell(\theta) = \frac{1}{2}(\theta - \theta^\star)^2$, which is $\mu$-PL and $L$-smooth with $\mu = L = 1$, and proxy gradient $\nabla\psi(\theta) = \alpha \nabla\ell(\theta) + b + \mathcal{N}(0, \sigma^2/B)$, where the fixed bias $b$ realizes the $(\alpha, \zeta)$-alignment of Assumption 2.1 through $\langle \nabla\psi, \nabla\ell \rangle \ge \alpha \|\nabla\ell\|^2 - \zeta$, and $B$ acts as the batch size via the gradient-noise variance $\sigma^2/B$. A distribution shift is simulated by moving $\theta^\star$ from 0 to $\Delta_W = 3$ with $\sigma = 3$. We set $\zeta = 10^{-3}$ and $\epsilon = 1$. We run the proxy-gradient baseline of Definition 3.8 and report the recovery time $\tau$ averaged over 100 runs.

### G.2. Controlled Synthetic Experiments

**Recovery complexity follows the predicted laws.** Table 1 reports $\tau$ as we vary the alignment strength $\alpha$: the product $\tau \cdot \alpha^2$ stays near-constant, confirming the predicted $\tau = \mathcal{O}(1/\alpha^2)$ scaling. Table 2 reports $\tau$ as we vary the batch size $B$: the product $\tau \cdot B$ stays near-constant, confirming the predicted $\tau = \mathcal{O}(1/B)$ scaling. Moreover, in both tables the measured $\tau$ stays above the lower bound (Theorem 3.3) and matches the order of the upper bound (Theorem 3.9) up to absorbed constants, empirically supporting the near-tightness of our matching bounds.

*Table 3.* Empirical proxy-task alignment $\tilde{\alpha}$ and Tent accuracy gain $\Delta$Acc., averaged over all 15 corruption types. The alignment is positive on every benchmark, and Tent improves on all corruptions.

| DATASET | $\tilde{\alpha}$ | $\Delta$ACC. | #IMPROVE |
|---|---|---|---|
| CIFAR-10-C | +0.066 | +22.96% | 15/15 |
| CIFAR-100-C | +0.705 | +14.13% | 15/15 |
| IMAGENET-C | +0.171 | +14.66% | 15/15 |

*Table 4.* ImageNet-C under Dirichlet temporal correlation with various $\beta$; smaller $\beta$ indicates stronger correlation. #Imp. measures the number of domains where Tent improves performance.

| $\beta$ | $\tilde{\alpha}$ | $\Delta$ACC. | #IMP. |
|---|---|---|---|
| 0.001 | $-0.028$ | $-12.65\%$ | 0 |
| 0.01 | +0.090 | +3.61% | 12 |
| 0.1 | +0.157 | +13.65% | 15 |
| UNIFORM | +0.171 | +14.66% | 15 |

## G.3. Real-World Experimental Setup

We evaluate the TTA baseline Tent (Wang et al., 2021) on the corruption benchmarks CIFAR-10-C, CIFAR-100-C, and ImageNet-C (Hendrycks & Dietterich, 2019). Our analysis reveals the proxy-task alignment $\alpha$ of Assumption 2.1 as the key quantity governing recovery. In the following experiments, we measure its empirical counterpart $\tilde{\alpha} = \langle \nabla \ell_{\text{ent}}, \nabla \ell_{\text{ce}} \rangle / \| \nabla \ell_{\text{ce}} \|^2$ in practice, which is the observed alignment between the entropy proxy gradient and the cross-entropy task gradient. $\tilde{\alpha} > 0$ indicates the positive supervision for TTA required by Assumption 2.1.

## G.4. Real-World Experiments

Table 3 reports $\tilde{\alpha}$ averaged over all 15 corruption types. The empirical alignment is positive on every benchmark, and Tent improves accuracy on all 15/15 corruptions of each dataset, with average gains of $+14\%$ to $+23\%$. Hence the alignment assumption underlying our analysis holds for a standard TTA algorithm across datasets of varying scale, and stronger alignment is associated with larger adaptation gains.

Table 4 further examines ImageNet-C streams generated by a Dirichlet distribution, where a smaller concentration $\beta$ induces stronger temporal correlation. Stronger temporal correlation lowers the empirical alignment $\tilde{\alpha}$: as $\beta$ decreases, $\tilde{\alpha}$ drops and eventually turns negative at $\beta = 0.001$, at which point Tent no longer improves any of the 15 corruptions and its accuracy degrades by 12.65%. The sign of $\tilde{\alpha}$ thus cleanly separates successful adaptation from collapse, providing a concrete empirical instance of our feasibility condition and a theoretical explanation for the model-collapse phenomenon reported in prior TTA studies.

