# OpenReview forum: "On the Learnability of Test-Time Adaptation: A Recovery Complexity Perspective"
_ICML.cc/2026/Conference — ICML 2026 regular_

### Official Review · Reviewer_Pwoy · 2026-02-25

**Soundness:** 2
**Presentation:** 2
**Significance:** 2
**Originality:** 3
**Overall Recommendation:** 3
**Confidence:** 4

**Summary:**

This paper investigates the under-explored topic of learnability in test-time adaptation (TTA) and proposes a novel theoretical framework based on Recovery Complexity, which quantifies the minimal time required for an algorithm to recover a target excess risk following a distribution shift in complex, non-stationary test streams.

**Compliance With Llm Reviewing Policy:**

Affirmed.

**Final Justification:**

The additional details in the rebuttal support the paper. Since the authors have addressed my concerns, I am increasing my rating to 3.

**Key Questions For Authors:**

Please refer to the weaknesses.

**Limitations:**

yes

**Strengths And Weaknesses:**

Strengths
1. This paper tackles a critical yet largely under-explored problem in test-time adaptation.
2. This paper provides a comprehensive theoretical framework, establishing order-wise matching minimax lower and upper bounds for recovery complexity. The in-depth analysis reveals the fundamental limits and the intrinsic adaptivity-information trade-off of TTA algorithms.

Weaknesses
1. The definition of the "non-stationary" test stream is unclear—it is ambiguous whether it specifically refers to a continual TTA setting or general Out-of-Distribution (OOD) data. Furthermore, the statement in Lines 222-223 ("A distribution shift occurs at time $t$=1, after which the distribution remains stationary") contradicts the core premise of a non-stationary stream. Additionally, the exact definition of "Recovered" is vague; does it imply achieving stable performance or avoiding model collapse?
2. The proposed theoretical framework relies on strong assumptions that appear to contradict well-known empirical phenomena in real-world TTA. The paper claims that an adapted model can successfully recover to a target excess risk within a bounded time complexity. However, existing literature [1] explicitly demonstrates that models often suffer from irreversible model collapse or severe catastrophic forgetting during continuous adaptation. Once collapsed, the performance is practically impossible to recover. This significant discrepancy between the theoretical guarantees and actual practical observations undermines the reliability of the proposed framework.
3. This paper lacks empirical evaluation, which makes it difficult to verify the practicality of the derived theoretical bounds or the validity of the proposed assumptions. The manuscript would be more convincing if empirical experiments were conducted on standard TTA benchmarks, such as ImageNet-C.

[1] Niu et al., Towards stable test-time adaptation in dynamic wild world, ICLR-2023.

---

> ### Author Rebuttal · Authors · 2026-03-29
>
> **Dear Reviewer Pwoy**:
>
> We thank the reviewer for the detailed and constructive feedback. We will address your concerns below.
>
> ---
> ## [W1] Definition of "Non-Stationary" and "Recovery"
>
> **Regarding “Non-Stationary”**:
> **We clarify that Lines 222-223 describe our local analysis within each approximated sub-stream, not a constraint on the full non-stationary stream.**
>
> Our paper studies continual TTA on a non-stationary test stream.
> Under Assumption 2.4, we approximate the full non-stationary stream by a sequence of piecewise-stationary sub-streams using Algorithm 1. Definition 3.1 defines recovery within each approximated sub-stream, and Theorem 4.3 transfers these local guarantees back to the original non-stationary stream.
> Therefore, the framework models continuously changing distributions, while using a standard piecewise-stationary reduction for tractable analysis.
> **We will revise our paper to make this clearer.**
>
> **Regarding "Recovery"**:
> **Recovery means that the excess risk falls below the target error threshold $\epsilon$ with probability at least $1-\delta$ within each approximated sub-stream.**
> Definition 3.1 then defines the recovery complexity as the minimum number of steps needed for this to happen.
> It is not a claim of stable performance under future shifts, and it is not merely a statement about avoiding collapse.
> When this target cannot be guaranteed, the method fails to recover, and model collapse is one type of failure.
> We further discuss this in W2 below.
>
> ---
> ## [W2] Connections to Model Collapse
>
> **We would like to clarify that our paper does not assume that recovery always occurs. On the contrary, it identifies when recovery cannot be guaranteed.**
>
> Remark 3.10 establishes a feasibility condition, i.e., $\zeta / (\alpha \cdot \mu) \leq \epsilon \cdot \delta/2$,
> under which recovery is achievable.
> **When this condition is violated, no TTA algorithm can guarantee $(\epsilon, \delta)$-recovery.**
>
> Therefore, existing empirical observations such as [1], including irreversible model collapse, do not contradict our theory.
> They fall into the part of our theory where recovery is not guaranteed because the proxy-task alignment is too weak.
> Our results are consistent with those observations and provide a concrete theoretical explanation for them.
> We provide further empirical support in W3 below.
>
> ---
> ## [W3] Empirical Evaluation on ImageNet-C
>
> Thank you for this valuable suggestion.
> We provide empirical validation on ImageNet-C using the Tent baseline, where test streams are generated via a Dirichlet distribution with parameter $\beta$, with smaller $\beta$ inducing stronger temporal correlation [2].
>
> In our paper, $\alpha$ measures proxy-task alignment. Empirically, we measure its counterpart
> $\tilde{\alpha} = \langle \nabla \ell_{\text{ent}}, \nabla \ell_{\text{ce}} \rangle / \|\nabla \ell_{\text{ce}}\|^2$
> on the current model and test stream, and report it together with the performance change ($\Delta$ Acc.) over 15 corruption types.
> We categorize each corruption as IMPROVE (accuracy exceeds baseline), DEGRADE (decreases but above 50% of baseline), or COLLAPSE (below 50% of baseline).
>
> |$\beta$|$\tilde{\alpha}$|$\Delta$ Acc.|#IMPROVE|#DEGRADE|#COLLAPSE|
> |-|-|-|-|-|-|
> |0.001|-0.0282|-12.65%|0|3|12|
> |0.01|+0.0901|+3.61%|12|3|0|
> |0.1|+0.1567|+13.65%|15|0|0|
> |Uniform|+0.1705|+14.66%|15|0|0|
>
> The results are consistent with our theoretical predictions.
> Under strong temporal correlation ($\beta=0.001$), the empirical alignment becomes negative, i.e., $\tilde{\alpha} < 0$, and most corruption types collapse.
> As $\tilde{\alpha}$ increases and becomes positive, adaptation becomes stable and yields performance improvement.
> **This supports our claim that the model collapse discussed in [1] does not contradict the theory.**
>
> We also ran controlled synthetic experiments to test the effects of alignment, batch size, and the feasibility condition, detailed in our responses to Reviewers fNwt and G3ZY.
> These experiments reproduce the predicted trends $\tau \propto 1/\alpha^2$ and $\tau \propto 1/B$ in Theorems 3.3 and 3.9, and they support the model collapse explanation in Remark 3.10.
>
> ---
> ## Reference
>
> [1] Towards stable test-time adaptation in dynamic wild world. ICLR 2023.
>
> [2] Note: Robust continual test-time adaptation against temporal correlation. NeurIPS 2022.

---

> > ### Author Rebuttal · Reviewer_Pwoy · 2026-04-02
> >
> > I appreciate the authors' effort in providing a detailed response.

---

> > > ### Author Response · Authors · 2026-04-02
> > >
> > > **Dear Reviewer Pwoy:**
> > >
> > > Thank you for your acknowledgment that the concerns have been fully resolved. We appreciate your careful consideration of our rebuttal.
> > >
> > > Given that the issues raised in the original review have now been addressed, we would like to ask if you would consider updating your overall recommendation to better reflect your current assessment of the paper.
> > >
> > > Thank you again for your time and consideration.
> > >
> > > Best regards,
> > >
> > > The Authors

---

### Official Review · Reviewer_h2Fg · 2026-03-06

**Soundness:** 2
**Presentation:** 2
**Significance:** 3
**Originality:** 3
**Overall Recommendation:** 4
**Confidence:** 2

**Summary:**

The paper proposes a new theoretical framework for TTA in recovery complexity and TTA learnability. Recovery complexity analyzes the theoretical time required for the TTA algorithm to achieve a target risk. TTA learnability measures the long-term reliability of the TTA algorithm. The paper also considers non-stationary / temporally-correlated streams to characterize the difficulty of TTA.

**Compliance With Llm Reviewing Policy:**

Affirmed.

**Final Justification:**

I believe this paper is an important step towards making TTA practical to real-world problems, so I maintain a score of 4.

**Key Questions For Authors:**

1. Would the Assumption 2.1 hold for entropy and cross-entropy objectives in practical TTA scenarios?

1. Can this theoretical analysis expand on complex TTA algorithms, including memory-based (e.g., NOTE, RoTTA, SoTTA) or teacher models (e.g., CoTTA, RoTTA)?

1. How can this theoretical analysis be applied to actual algorithms/evaluations/problem settings?

**Limitations:**

yes

**Strengths And Weaknesses:**

Strenghts

1. The paper targets the underexplored field of the theoretical framework of TTA.

1. The paper targets a wide range of TTA scenarios, including online, continual, and temporally-correlated settings.


Weaknesses

1. The paper relies on heavy assumptions and various parameters. For example, it would be better to empirically explore whether Assumption 2.1 holds for typical Entropy/Cross-Entropy losses.

1. The paper is purely theoretical, limiting its practical validation. This questions the importance of the theoretical analysis.

1. Although it is reasonable to analyze one-step stochastic gradient TTA algorithms, many TTA algorithms are beyond the stochastic gradient descent, including: memory structures (e.g., NOTE, RoTTA, SoTTA) and teacher models (e.g., CoTTA, RoTTA).



Minor points

1. Please remove the duplicated references.

---

> ### Author Rebuttal · Authors · 2026-03-29
>
> **Dear Reviewer h2Fg**:
>
> We thank the reviewer for the constructive feedback.
>
> ---
> ## [W1&Q1&W2] Assumption 2.1 and Empirical Validation
>
> Thank you for your suggestion.
>
> **Assumption 2.1 requires that the proxy gradient provides a useful descent direction for the task loss**, i.e., a positive alignment between the gradients of entropy minimization and cross-entropy losses in classical TTA algorithms.
>
> To empirically examine this, in our paper $\alpha$ measures proxy-task alignment, and we use
> $\tilde{\alpha} = \langle \nabla \ell_{\text{ent}}, \nabla \ell_{\text{ce}} \rangle / \|\nabla \ell_{\text{ce}}\|^2$
> as its empirical counterpart under the Tent baseline, averaged across corruption types. Thus, positive $\tilde{\alpha}$ indicates positive empirical proxy-task alignment, consistent with the role of $\alpha$ in Assumption 2.1.
>
> | Dataset | $\tilde{\alpha}$ | $\Delta$ Acc. | #IMPROVE |
> |-|-|-|-|
> | CIFAR-10-C  | +0.0656 | +22.96% | 15/15 |
> | CIFAR-100-C  | +0.7047 | +14.13% | 15/15 |
> | ImageNet-C  | +0.1705 | +14.66% | 15/15 |
>
> **The average $\tilde{\alpha} > 0$ across all three datasets**, showing that the entropy proxy gradient is empirically positively aligned with the task gradient. On CIFAR-10-C, Spearman $\rho(\tilde{\alpha}, \Delta\text{Acc}) = 0.60$, suggesting that stronger alignment is associated with larger adaptation gains.
>
> Conversely, in our response to Reviewer Pwoy, we show that when $\tilde{\alpha} < 0$ (induced by temporal correlation), TTA algorithms tend to fail, further supporting the role of alignment as an important factor.
> We also provide controlled experiments to verify our conclusions $\tau \propto 1/\alpha^2$ and $\tau \propto 1/B$. Please refer to our response to Reviewer fNwt and Reviewer G3ZY for detailed results due to the character limits.
>
> **We will restate Assumption 2.1 to hold locally within $\mathcal{N}_r(\theta_1)$ rather than globally**, making it consistent with the local nature of other assumptions and better aligned with practical TTA algorithms.
>
> ---
> ## [W3&Q2] Extending Theoretical Results to TTA algorithms
>
> Thank you for this important question.
>
> Our paper is formulated at the problem level and does not assume a specific algorithm. **Specifically, the minimax lower bound (Theorem 3.3) is algorithm-independent, and applies to all TTA algorithms**, including NOTE, CoTTA, and so on. It measures the challenge of TTA under unlabeled non-stationarity, regardless of the algorithm used.
>
> The upper bound (Theorem 3.9) shows that even a simple proxy-SGD already achieves near-optimal recovery. This suggests that advanced TTA algorithms are still subject to the same problem-level limits, but can be understood as improving the problem parameters, such as alignment ($\alpha$, $\zeta$) and temporal dependence ($C_\phi$).
> For example, **NOTE uses a memory bank to balance label distribution, which mitigates temporal correlation and effectively reduces $C_\phi$ (Proposition 2.9)**,
> while **CoTTA uses a teacher-augmented soft-label proxy that is better aligned with the task loss**, improving $\tilde{\alpha}$ from 0.0656 (Tent) to 0.4741 on CIFAR-10-C.
> RoTTA combines memory bank and teacher model, simultaneously reducing $C_\phi$ and improving $\alpha$.
> SoTTA filters low-confidence samples, effectively raising the average $\alpha$.
>
> Therefore, our framework helps interpret diverse TTA algorithms through a common set of problem parameters.
> We will add a discussion of how existing TTA algorithms map to our theoretical parameters in our revision.
>
> ---
> ## [W2&Q3] Practical Applications
>
> Thank you for this important question. Our framework offers applications in algorithms, evaluations, and problem settings.
>
> **Algorithms:** Our bounds show $\tau \propto 1/\alpha^2$ but $\tau \propto C_\phi/B$, meaning improving proxy-task alignment ($\alpha$) yields quadratic gains, while mitigating temporal correlation ($C_\phi$) yields only linear gains.
> This quantifies the relative impact of each factor, helping identify the bottleneck in a given TTA problem.
>
> **Evaluations:** Standard TTA benchmarks report a single accuracy. Our framework decomposes performance into several key factors, enabling researchers to verify different TTA mechanisms and their contributions under controlled settings. We demonstrate this in our response to Reviewers fNwt and G3ZY.
>
> **Problem settings:** Our minimax lower bound (Theorem 3.3) characterizes the fundamental difficulty of a TTA problem through its parameters. This helps researchers distinguish whether poor performance is due to a hard problem or a suboptimal algorithm, and suggests how to improve the algorithm.
>
> ---
> ## Minor: Duplicated References
>
> Thank you for pointing this out. We will fix the duplicated references in our revision.

---

> > ### Author Rebuttal · Reviewer_h2Fg · 2026-04-05
> >
> > I acknowledge that the author rebuttal has fully addressed my concerns. I believe this paper is an important step towards making TTA practical to real-world problems.

---

> > > ### Author Response · Authors · 2026-04-06
> > >
> > > **Dear Reviewer h2Fg**:
> > >
> > > Thank you for your positive feedback and for acknowledging that our rebuttal has addressed your concerns.
> > >
> > > We appreciate your recognition of the importance of this work. We will further improve the paper in our final version to enhance the clarity and completeness.
> > >
> > > Thank you again for your time and consideration.
> > >
> > > Best regards,
> > >
> > > The Authors

---

### Official Review · Reviewer_fNwt · 2026-03-09

**Soundness:** 2
**Presentation:** 3
**Significance:** 3
**Originality:** 3
**Overall Recommendation:** 4
**Confidence:** 5

**Summary:**

This paper develops a theory-first framework for test-time adaptation (TTA) under unlabeled, non-stationary test streams. Its main contribution is to replace regret-centric analysis with two TTA-specific notions: $(\epsilon, \delta)$-recovery complexity, measuring how long an algorithm needs to recover after a shift, and $(\epsilon, \rho)$-TTA learnability, measuring the long-run fraction of time steps on which excess error exceeds $\epsilon$. The paper combines a quantized Wasserstein model for evolving distributions, a $\phi$-mixing model for temporal dependence, and a proxy-optimal comparator to derive a minimax lower bound, an upper bound for a simple one-step proxy-SGD baseline, and a transfer from local recovery guarantees to global learnability and dynamic regret. The overall framing is novel and relevant to modern TTA.

**Compliance With Llm Reviewing Policy:**

Affirmed.

**Final Justification:**

The rebuttal partially addresses my concerns, but it does not fully resolve the central issue. Regarding the lower-bound proof, the authors clarified that Eq. (30) is a constructed hard instance for the minimax lower bound rather than something derived from Assumption 2.1; if this is made explicit in the final version, I consider that concern largely addressed. The added synthetic and real-world experiments in the rebuttal are also helpful and improve the paper’s completeness.

However, my concern remains the strength of the proxy-task alignment assumption. The authors acknowledge that the original globally stated alignment assumption is too strong and propose to localize Assumption 2.1 to a neighborhood, but this feels more like a clarification than a substantive weakening of the theory. The rebuttal still does not sufficiently explain why this local assumption is realistic in practical TTA settings, nor does it fully address the limited scope of applicability of the theoretical results.

Overall, the rebuttal improves clarity and completeness, so I'm adjusting my score accordinly.

**Key Questions For Authors:**

Please refer to the weakness section.

**Limitations:**

* The main limitation is the global alignment assumption which is difficult to justify in large nonconvex models. In practice, proxy and task gradients are likely aligned only locally near initialization or along the adaptation trajectory.
* The paper explicitly notes that it does not evaluate the tightness of constants or the practical impact of the assumptions. Given the strength of the assumptions, this is a significant limitation.

**Strengths And Weaknesses:**

Strength:

* The paper identifies a real mismatch between dynamic regret and the operational goals of TTA.
* Recovery complexity and TTA learnability are useful and natural problem formulations.
* The split between global drift, local dependence, and proxy alignment is analytically appealing.
* The lower-bound / upper-bound / local-to-global chain is ambitious and, at a high level, well designed.

Weakness:

* The main alignment assumption seems too strong. I think the central assumption is hard to justify in realistic settings. The paper assumes a uniform $(\alpha, \zeta)$ condition for all parameters in $\Theta$, which means the proxy gradient must stay aligned with the task gradient everywhere. Maybe this is reasonable in a very structured model, but in large nonconvex spaces it seems easy to find regions where the two gradients are weakly aligned or even point in opposite directions.
* I also think there is a mismatch in the setup: the paper only uses local smoothness/PL/Lipschitz conditions in a neighborhood, but alignment is assumed globally. Maybe it would be more believable to assume alignment only locally, or only along the algorithm path. As written, I think the results are better read as theory for globally aligned proxy models, not for generic TTA.
* I do not think the lower-bound proof matches the stated theorem. Theorem 3.3 is stated under Assumptions 2.1, 2.4, and 2.7. But I think the appendix proof seems to use something stronger than Assumption 2.1. In particular, the proof appears to rely on an additive decomposition like $\nabla \psi = \alpha \nabla \ell + \xi$, or an equivalent bias model, while Assumption 2.1 only gives an inner-product lower bound.
* There is no empirical validation. I think this is a real weakness. Since the theory depends on strong assumptions, I would have liked at least a small empirical section checking whether those assumptions hold even roughly in practice. For example, maybe the paper could measure alignment, temporal dependence, or how recovery time changes with $B$, $\alpha$, or $\zeta$. Without that, it is hard to tell whether the framework captures an interesting part of TTA practice or mainly a clean abstraction.

---

> ### Author Rebuttal · Authors · 2026-03-29
>
> **Dear Reviewer fNwt**:
>
> We thank the reviewer for the valuable and technically substantive feedback.
>
> ---
> ## [W1&W2&L1] Alignment Assumption
>
> We agree that requiring a global alignment over the entire parameter space can be overly strong, and that the current presentation creates a mismatch between the globally stated alignment assumption and the otherwise local assumptions.
>
> In particular, our analysis focuses on recovery in a local neighborhood, which is already consistent with Assumption 2.2 and the local comparator in Definition 2.11.
> **In the revision, we will restate Assumption 2.1 on $\mathcal{N}_r(\theta_1)$ and make this locality explicit in the related statements and proof presentation.**
> This is mainly a clarification of the assumption so that it matches the local structure already used elsewhere in the paper.
>
> We appreciate this suggestion, which improves the consistency of the assumptions and makes the paper better aligned with practical TTA.
>
> ---
> ## [W3] Lower-Bound Proof
>
> **Eq.(30) is not derived from Assumption 2.1, but is a constructed hard instance for the minimax lower bound.** We then verify that this construction satisfies Assumption 2.1, which reduces to the constraint Eq.(33) on the bias $\xi$, and choose parameters accordingly using Eq.(38). The additive structure in Eq.(30) is a design choice for the hard instance in the proof, not an additional modeling assumption.
>
> **We will revise the appendix to make explicit that Eq.(30) is a construction and that Eqs.(31)–(33) serve as its verification against Assumption 2.1, yielding the constraint on $\xi$.**
> We will also carefully check and improve the presentation of the theory and proofs.
>
> ---
> ## [W4&L2] Empirical Validation
>
> We provide controlled synthetic experiments to verify the predicted relations and feasibility condition (EXP 1-3), and real-world experiments to validate our theorems in practice (EXP 4).
>
> For EXP 1-3, we use a 1D quadratic loss $\ell(\theta)=\frac{1}{2}(\theta-\theta^\*)^2$ with proxy gradient $\nabla \psi(\theta)=\alpha(\theta-\theta^\*) + \xi + \mathcal{N}(0,\sigma^2/B)$, where $\alpha$ controls alignment.
> $\xi$ induces bias with $\|\xi\|^2/\alpha \leq \zeta$ as in Eq.(38).
> $B$ controls the gradient noise variance, which corresponds to batch size.
> We simulate a shift ($\theta^*$: $0 \to 3$, $\sigma=3.0$) and measure $\tau$ over 100 runs.
>
> **`EXP 1`: $\tau$ vs $\alpha$**
>
> Fix $B=16$, $\zeta=0.001$, $\epsilon=1.0$, vary $\alpha$.
>
> | $\alpha$ | 0.05 | 0.10 | 0.20 | 0.30 |
> |-|-|-|-|-|
> | $\tau$ | 322.0 | 77.0 | 19.0 | 9.2 |
> | $\tau \cdot \alpha^2$ | 0.80 | 0.77 | 0.76 | 0.83 |
>
> $\tau \cdot \alpha^2 \approx$ const, supporting $\tau \propto 1/\alpha^2$ (Theorems 3.3, 3.9).
>
> **`EXP 2`: $\tau$ vs $B$**
>
> Fix $\alpha=0.2$, $\zeta=0.001$, $\epsilon=1.0$, vary $B$.
>
> | $B$ | 1 | 4 | 16 | 32 |
> |-|-|-|-|-|
> | $\tau$ | 323.5 | 82.0 | 19.0 | 10.6 |
> | $\tau \cdot B$ | 323.5 | 328.0 | 304.0 | 339.2 |
>
> $\tau \cdot B \approx$ const, supporting $\tau \propto 1/B$ (Theorems 3.3, 3.9).
>
> **`EXP 3`: Feasibility Condition**
>
> Vary $\zeta/\alpha$ with $\alpha=0.5$, $B=64$, $\epsilon=0.5$, $\delta=0.1$, $\mu=1$.
> The theoretical feasibility threshold from Remark 3.10 is $\zeta/(\alpha\mu) \leq \epsilon\delta/2 = 0.025$.
>
> | $\zeta/\alpha$ | 0.010 | 0.025 | 0.500 | 1.000 | 2.000 | 5.000 |
> |-|-|-|-|-|-|-|
> | Recovery | 100% | 100% | 99% | 86% | 48% | 5% |
>
> Recovery is 100% when $\zeta/\alpha \leq 0.025$, and becomes increasingly unreliable as the violation grows, supporting the feasibility condition in Remark 3.10.
>
> **`EXP 4`: Real-World Alignment Verification**
>
> In our paper, $\alpha$ measures proxy-task alignment. We use
> $\tilde{\alpha} = \langle \nabla \ell_{\text{ent}}, \nabla \ell_{\text{ce}} \rangle / \|\nabla \ell_{\text{ce}}\|^2$ as its empirical counterpart. Thus, positive $\tilde{\alpha}$ indicates positive empirical proxy-task alignment, consistent with the role of $\alpha$ in Assumption 2.1.
> We measure $\tilde{\alpha}$ across three benchmarks with Tent, averaged over all 15 corruptions.
>
> | Dataset | $\tilde{\alpha}$ | $\Delta Acc.$ | #IMPROVE |
> |-|-|-|-|
> | CIFAR-10-C | +0.0656 | +22.96% | 15/15 |
> | CIFAR-100-C | +0.7047 | +14.13% | 15/15 |
> | ImageNet-C | +0.1705 | +14.66% | 15/15 |
>
> The average $\tilde{\alpha} > 0$ across all three datasets, providing empirical support for positive proxy-task alignment in practice.
>
> Moreover, in our response to Reviewer Pwoy, we show that on ImageNet-C with Dirichlet-controlled temporal correlation ($\beta=0.001$), $\tilde{\alpha}$ turns negative ($-0.028$) and Tent collapses ($\Delta Acc. =-12.65$%), while $\beta \geq 0.1$ yields $\tilde{\alpha}>0$ and the model can recover. This supports that alignment is an important factor for TTA success on real-world data.

---

> > ### Author Rebuttal · Reviewer_fNwt · 2026-04-03
> >
> > The rebuttal partially addresses my concerns, but it does not fully resolve the central issue. Regarding the lower-bound proof, the authors clarified that Eq. (30) is a constructed hard instance for the minimax lower bound rather than something derived from Assumption 2.1; if this is made explicit in the final version, I consider that concern largely addressed. The added synthetic and real-world experiments in the rebuttal are also helpful and improve the paper’s completeness.
> >
> > However, my concern remains the strength of the proxy-task alignment assumption. The authors acknowledge that the original globally stated alignment assumption is too strong and propose to localize Assumption 2.1 to a neighborhood, but this feels more like a clarification than a substantive weakening of the theory. The rebuttal still does not sufficiently explain why this local assumption is realistic in practical TTA settings, nor does it fully address the limited scope of applicability of the theoretical results.
> >
> > Overall, the rebuttal improves clarity and completeness, so I'm adjusting my score accordinly.

---

> > > ### Author Response · Authors · 2026-04-04
> > >
> > > **Dear Reviewer fNwt,**
> > >
> > > Thank you again for the thoughtful follow-up and for updating your evaluation. We truly appreciate the careful reading and the technically insightful feedback. Your comments on the alignment assumption have significantly improved the clarity and positioning of our paper.
> > >
> > > We understand your remaining concern regarding the connection between our theoretical assumptions and practical TTA settings.
> > > Our paper does not assume that proxy-task alignment holds broadly, but rather focuses on the situations where TTA can succeed. In practice, many TTA algorithms operate under constrained updates (e.g., partial parameter adaptation or implicit regularization), which tend to keep the model close to the initialization. Our local formulation is intended to reflect this.
> > >
> > > We also agree that this condition does not cover all possible TTA scenarios.
> > > We will make this scope more explicit in the final version, clarifying that our results identify conditions under which TTA can succeed, rather than providing a universal guarantee for all TTA problems.
> > >
> > > Thank you again for your time and consideration.
> > >
> > > Best regards,
> > >
> > > The Authors

---

### Official Review · Reviewer_G3ZY · 2026-03-12

**Soundness:** 3
**Presentation:** 4
**Significance:** 4
**Originality:** 4
**Overall Recommendation:** 5
**Confidence:** 2

**Summary:**

The paper proposes a theoretical framework for understanding when test-time adaptation (TTA) is possible under non-stationary test data streams. The authors address the lack of theory for TTA by introducing two key concepts: Recovery Complexity, which measures how quickly a model recovers to a target accuracy after a distribution shift, and learnability, which evaluates the long-term reliability of TTA algorithms across an evolving test stream. To model real-world non-stationarity, they propose a unified stream formulation that captures both distribution shifts and temporal correlations in the data. Next, they discuss minimax lower bounds and matching upper bounds on recovery complexity, revealing fundamental limits and trade-offs in TTA, including the impact of proxy-task misalignment and batch size.

**Compliance With Llm Reviewing Policy:**

Affirmed.

**Final Justification:**

Thanks to the authors for their efforts. I would like to keep my initial score of accepting the paper.

**Key Questions For Authors:**

Questions:
- The paper assumes that the learning setting is composed of proxy–task gradient alignment, smoothness, and PL conditions, bounded variance gradients.
- The work mainly analyzes a simple gradient-based adaptation baseline, without proposing new practical algorithms. Therefore, its impact may remain mostly conceptual.
- How would the scenario change, especially for the formulation of distribution shifts (Section 2.2), when there is the presence of multimodal distribution shifts? These things can involve multiple distribution shifts and the goal of adaptation meanwhile is to adapt to only one of them and ignore the rest

**Limitations:**

Yes.

**Strengths And Weaknesses:**

**Strengths**
- Novel theory work for Test-time adaptation with Recovery Complexity. And first learnability theory framework for test-time adaptation
- Unified modelling of distribution shifts with gradual and abrupt distribution shifts using Wasserstein-based quantization

**Weaknesses**
Overall, it's a theory paper with no experiments or empirical evidence, which is the only weakness I could point out.
I have few questions about limiting their study to few distribution shifts and not incuding multi-modal distribution shifts

---

> ### Author Rebuttal · Authors · 2026-03-29
>
> **Dear Reviewer G3ZY**:
>
> We thank the reviewer for the positive assessment and thoughtful questions.
>
> ---
> ## [Q1&W1] Empirical Validation
>
> Thank you for your valuable suggestions.
>
> These assumptions are standard in optimization-based analysis, and we have empirically validated the key alignment assumption on real-world benchmarks (EXP 1). We will restate Assumption 2.1 to hold locally within $N_r(\theta_1)$ rather than globally, making it consistent with the local nature of other assumptions and better aligned with practical TTA algorithms.
>
> Moreover, we have conducted the following empirical experiments to verify our theoretical results.
>
> **`EXP 1`: Alignment Assumption on Real-World Datasets**
>
> In our paper, $\alpha$ measures proxy-task alignment. We use
> $\tilde{\alpha} = \langle \nabla \ell_{\text{ent}}, \nabla \ell_{\text{ce}} \rangle / \|\nabla \ell_{\text{ce}}\|^2$
> as its empirical counterpart, measured on the current model and test stream using labeled test data for validation purposes across three benchmarks with the Tent baseline. Thus, positive $\tilde{\alpha}$ indicates positive empirical proxy-task alignment, consistent with the role of $\alpha$ in Assumption 2.1.
>
> | Dataset | $\tilde{\alpha}$ | $\Delta$ Acc. | #IMPROVE |
> |-|-|-|-|
> | CIFAR-10-C  | +0.0656 | +22.96% | 15/15 |
> | CIFAR-100-C  | +0.7047 | +14.13% | 15/15 |
> | ImageNet-C  | +0.1705 | +14.66% | 15/15 |
>
> The average $\tilde{\alpha} > 0$ across all three datasets, providing empirical support for positive proxy-task alignment in practice. Stronger alignment is also generally associated with better adaptation performance in these TTA settings.
> Moreover, in our response "W3: Empirical Evaluation on ImageNet-C" to Reviewer Pwoy, we also show that when $\tilde{\alpha} < 0$, TTA algorithms tend to fail.
>
> **`EXP 2`: Controlled Synthetic Experiments**
>
> We have presented controlled synthetic experiments to verify $\tau \propto 1/\alpha^2$, $\tau \propto 1/B$, and the feasibility condition. Please refer to "W4 & L2: Empirical Validation" in our response to Reviewer fNwt for detailed results.
>
> Moreover, we also verify the bound tightness across 7 experiment configurations:
>
> | $\alpha$ | $B$ | LB | $\tau$ | UB  |
> |-|-|-|-|-|
> | 0.05 | 16 | 59.0 | 322.0 | 311.9 |
> | 0.10 | 16 | 15.6 | 77.0 | 78.0 |
> | 0.20 | 16 | 4.1 | 19.0 | 19.5 |
> | 0.50 | 16 | 0.7 | 4.0 | 3.1 |
> | 0.20 | 1 | 65.2 | 323.5 | 311.9 |
> | 0.20 | 4 | 16.3 | 82.0 | 78.0 |
> | 0.20 | 64 | 1.0 | 6.0 | 4.9 |
>
> The empirical $\tau$ is close to the predicted upper bound across all configurations, supporting the near-tightness of our minimax bounds.
> In several cases, $\tau$ slightly exceeds the UB, which is expected since both bounds are order-wise and the upper bound omits universal constants.
>
> ---
> ## [W1&Q3] Multi-modal Distribution Shifts
>
> Thank you for raising this interesting scenario.
>
> When adapting to the overall distribution, our framework applies directly. Multi-modal shifts increase effective non-stationarity by mixing gradient signals, leading to larger $V_T$ and more segments $\tilde{K}_S(T)$, and thus a larger $\rho$ in Theorem 4.3.
>
> For the case where the goal is to adapt to only one mode of multi-modal distribution shifts, the problem fundamentally changes from adapting to $P_t$ to selecting a latent component of a mixture. The key challenge becomes mode identification from unlabeled and non-stationary data.
> Once a mode is identified, adaptation reduces to a conditional sub-stream where our recovery analysis (Theorems 3.3, 3.9) applies. Extending the framework to jointly handle identification and recovery is an important direction for future work.
>
> ---
> ## [Q2] Practical Impact
>
> We appreciate this observation.
> Our framework offers concrete practical impact beyond the theoretical contributions.
> Since the bounds reveal $\tau \propto 1/\alpha^2$ and $\tau \propto C_\phi / B$, it helps explain why several existing TTA mechanisms are effective.
> For example, NOTE[1] uses a memory bank to balance label distribution, mitigating temporal correlation and effectively reducing $C_\phi$ (Proposition 2.9).
> CoTTA[2] uses a teacher-augmented soft-label proxy that is better aligned with the task loss, improving $\tilde{\alpha}$ from 0.0656 (Tent) to 0.4741 on CIFAR-10-C.
>
> Moreover, the minimax lower bound (Theorem 3.3) is algorithm-independent, revealing the difficulty of a TTA problem. This helps researchers distinguish whether poor performance is due to a hard problem or a suboptimal algorithm, and suggests how to improve the algorithm.
>
> Our paper complements existing empirical works by clarifying what aspects of TTA can be improved algorithmically and what are fundamentally constrained.
>
> ---
> ## Reference
>
> [1] Note: Robust continual test-time adaptation against temporal correlation. NeurIPS 2022.
>
> [2] Continual test-time domain adaptation. CVPR 2022.

---

> > ### Author Rebuttal · Reviewer_G3ZY · 2026-04-04
> >
> > Thanks for the efforts. The questions I had, though not major, have been clarified. Thanks for providing the detailed experiments that were missing from the paper. I encourage you to include them in the paper and also state it in the main paper.

---

> > > ### Author Response · Authors · 2026-04-04
> > >
> > > **Dear Reviewer G3ZY:**
> > >
> > > Thank you for the positive feedback and for acknowledging that your concerns have been resolved. We truly appreciate your careful reading and constructive suggestions.
> > >
> > > We will incorporate the experiments into the main paper and clearly present them to strengthen the empirical support of our work.
> > >
> > > Thank you again for your time and consideration.
> > >
> > > Best regards,
> > >
> > > The Authors

---

### Decision · Program_Chairs · 2026-04-30

**Decision:**

Accept (regular)

**Comment:**

Great paper! Post rebuttal, all reviewers have found the clarifications of the novelty of results and additional experiments performed helpful. The novelty of the framework has been on firmer ground with the lower bound proof clarification, and the additional experiments further validated theory and provided additional empirical support. The rebuttal comments are clearly formulated, and in my view has turned the more negative reviewers around in the value of the paper (while still keeping the positive sentiment of the remaining reviewers).